# ALIGNING VISUAL CONTRASTIVE LEARNING MODELS VIA PREFERENCE OPTIMIZATION

**Amirabbas Afzali & Borna Khodabandeh** [*]
{amir8afzali, borna710kh}@gmail.com

**Ali Rasekh**
L3S Research Center,
Leibniz Universität Hannover, Germany
ali.rasekh@L3S.de

**Mahyar JafariNodeh**
Massachusetts Institute of Technology, USA
mahyarjn@mit.edu

**Sepehr Kazemi**
sepehrkazemi9@gmail.com

**Simon Gottschalk**
L3S Research Center,
Leibniz Universität Hannover, Germany
gottschalk@L3S.de

## ABSTRACT

Contrastive learning models have demonstrated impressive abilities to capture semantic similarities by aligning representations in the embedding space. However, their performance can be limited by the quality of the training data and its inherent biases. While Preference Optimization (PO) methods such as Reinforcement Learning from Human Feedback (RLHF) and Direct Preference Optimization (DPO) have been applied to align generative models with human preferences, their use in contrastive learning has yet to be explored. This paper introduces a novel method for training contrastive learning models using different PO methods to break down complex concepts. Our method systematically aligns model behavior with desired preferences, enhancing performance on the targeted task. In particular, we focus on enhancing model robustness against typographic attacks and inductive biases, commonly seen in contrastive vision-language models like CLIP. Our experiments[1] demonstrate that models trained using PO outperform standard contrastive learning techniques while retaining their ability to handle adversarial challenges and maintain accuracy on other downstream tasks. This makes our method well-suited for tasks requiring fairness, robustness, and alignment with specific preferences. We evaluate our method for tackling typographic attacks on images and explore its ability to disentangle gender concepts and mitigate gender bias, showcasing the versatility of our approach.

## 1 INTRODUCTION

In recent years, Vision-Language Models (VLMs) such as CLIP (Radford et al., 2021) have revolutionized downstream vision-language tasks like classification (Conde & Turgutlu, 2021), object detection (Zhong et al., 2021), segmentation (Xu et al., 2022), and image generation (Saharia et al., 2022). These models are trained on large-scale web data, such as the 400 million text-image pairs used for training CLIP. However, despite their success, VLMs exhibit several vulnerabilities. For instance, adversarial attacks (Goodfellow et al., 2015; Moosavi-Dezfooli et al., 2016) and backdoor vulnerabilities (Bai et al., 2024; Liang et al., 2024a) can lead to erroneous classifications with high confidence. CLIP has also been shown to be vulnerable to typographic attacks, where text within images causes misclassification (Goh et al., 2021). Additionally, biases such as gender and racial bias

---

[*]Equal contribution.
[1]The code is available on GitHub.

(Ruggeri & Nozza, 2023; Zhang et al., 2022; Bolukbasi et al., 2016; Jialu Wang et al., 2021) can be amplified by these models due to biases in their training datasets. Another dispreferred behavior occurs when models focus on tasks unrelated to the intended objective, as highlighted by (Menon et al., 2022). These challenges underscore the need for alignment methods in large pre-trained models.

Generative models, including Large Language Models (LLMs), have also been widely adopted for solving complex problems across various domains. Examples in the literature include GPT-4 (Achiam et al., 2023), Mistral (Albert Q. Jiang et al., 2023), and LLaMA (Touvron et al., 2023). These models are trained on massive datasets of unlabeled text, learning general language representations. Some of these LLMs require additional Supervised Fine-Tuning (SFT) to specialize for specific tasks using labeled data. However, even with fine-tuning, LLMs can produce outputs that misalign with human values or safety expectations, such as the unreliability of LLMs in autonomous driving (Chen et al., 2024). To mitigate this, alignment techniques are applied, where models are further trained using a reward model that captures human preferences (Stiennon et al., 2022; Ouyang et al., 2022; Bai et al., 2022). Popular alignment approaches include Reinforcement Learning from Human Feedback (RLHF) (Christiano et al., 2023) and Direct Preference Optimization (DPO) (Rafailov et al., 2024), both of which extend the capabilities of these models beyond what SFT alone can achieve. These alignment techniques have improved the safety and relevance of LLM outputs by making them more aligned with human preferences.

Alignment methods have been explored in VLMs (Sun et al., 2023; Gallego, 2023; Clark et al., 2023), but to the best of our knowledge, they have not been extensively applied to contrastive learning models like CLIP. In this work, we extend the preference optimization paradigm to non-generative VLMs. Our approach builds on the contrastive learning framework while simultaneously aiming to preserve the pretrained knowledge—a concept known as continual learning (Garg et al., 2024; Wang et al., 2023).

In this work, we extend the preference optimization paradigm to contrastive learning models, exemplified on enhancing robustness against typographic attacks and mitigating gender biases in image classification and retrieval. By systematically aligning the model's behavior with human preferences, we aim to preserve its pretrained knowledge while improving its performance in sensitive areas such as fairness and robustness. The main contributions of this paper are as follows:

- We extend and compare three PO methods, originally developed for aligning generative models, to non-generative contrastive learning models. This way, we improve model alignment, preserving the pretrained model knowledge while optimizing for desired behaviors.
- We propose controlling model behavior by adjusting the singular values of a learnable linear transformation. This allows fine-tuning regarding specific concepts.
- Our experiments demonstrate the benefits of applying preference optimization in contrastive learning tasks, such as training for robustness against typographic attacks and disentangling gender information from the embedding space, mitigating gender bias.

## 2 FOUNDATIONS & RELATED WORK

**Preference Optimization and RLHF.** Reinforcement Learning from Human Feedback (RLHF) aligns model behavior with human preferences by training the model based on feedback from human annotators. In the standard RLHF paradigm, annotators rank model responses. For example, given an input prompt $x$, a preference inequality might be expressed as $y_l \prec y_w$, meaning response $y_w$ is preferred over response $y_l$ for prompt $x$. These preferences are structured into datasets $\mathcal{D} = \{(x, y_w, y_l)\}$. The preference modeling often follows energy-based methods like the Bradley-Terry model (Bradley & Terry, 1952), where the probability of preferring $y_w$ over $y_l$ is modeled as below, where $r^*$ is the "true" reward function underlying the preferences:

$$P^*(y_w \succ y_l \mid x) = \frac{e^{r^*(x,y_w)}}{e^{r^*(x,y_w)} + e^{r^*(x,y_l)}} = \sigma\left(r^*(x, y_w) - r^*(x, y_l)\right). \tag{1}$$

Since obtaining the true reward $r^*$ from a human is impossible, a reward model $r_\phi$ is learned by approximating the true reward function. This model is optimized by minimizing the negative log-likelihood of the human preference data:

$$\mathcal{L}_R(r_\phi) = \mathbb{E}_{x,y_w,y_l \sim \mathcal{D}}[-\log(\sigma(r_\phi(x, y_w) - r_\phi(x, y_l)))]. \tag{2}$$

Once the reward function is learned, the RL pipeline can be used to improve the generation policy. To prevent significant deviation from the pretrained reference model, a KL-divergence-shaped reward is incorporated, with the RL objective defined as:

$$\max_{\pi_\theta} \mathbb{E}_{x \sim \mathcal{D}, y \sim \pi_\theta}[r_\phi(x, y)] - \beta D_{\mathrm{KL}}(\pi_\theta(y|x) \| \pi_{\mathrm{ref}}(y|x)), \tag{3}$$

where $\beta > 0$ controls the degree of allowed divergence from $\pi_{\mathrm{ref}}$. Unlike RLHF, which is often slow and unstable due to its reliance on sampling and reinforcement learning, DPO (Rafailov et al., 2024) offers a more efficient and stable alternative by directly defining a loss function and bypassing reinforcement learning, allowing for standard optimization. However, DPO can sometimes overfit to preference datasets. To address this, Identity Preference Optimization (IPO) (Gheshlaghi Azar et al., 2024) bypasses the Bradley-Terry modelization and introduces a new loss function to decrease overfitting by controlling the gap between the likelihood ratios of the model and a reference model. Defining the *policy ratio* as below:

$$h_{\pi_\theta}(y_w, y_l; x) = \log\left(\frac{\pi_\theta(y_w|x)\pi_{\mathrm{ref}}(y_l|x)}{\pi_\theta(y_l|x)\pi_{\mathrm{ref}}(y_w|x)}\right). \tag{4}$$

We can now express both the DPO and IPO objectives in terms of this ratio:

$$\mathcal{L}_{\mathrm{DPO}}(\pi_\theta, \pi_{ref}) = \mathbb{E}_{(x, y_l, y_w) \sim \mathcal{D}}\left[-\log \sigma\left(\beta h_{\pi_\theta}(y_w, y_l, x)\right)\right], \tag{5}$$

$$\mathcal{L}_{\mathrm{IPO}}(\pi_\theta; \pi_{\mathrm{ref}}) = \mathbb{E}_{(x, y_w, y_l) \sim \mathcal{D}}\left[\left(h_{\pi_\theta}(y_w, y_l; x) - \frac{\beta^{-1}}{2}\right)^2\right]. \tag{6}$$

Additionally, newer methods like Kahneman-Tversky Optimization (KTO) (Ethayarajh et al., 2024) only require a binary signal of whether an output is desired or undesired, making it more practical for many real-world applications:

$$\mathcal{L}_{\mathrm{KTO}}(\pi_\theta, \pi_{\mathrm{ref}}) = \mathbb{E}_{x, y \sim \mathcal{D}}[w(y)(1 - v_{\mathrm{KTO}}(x, y; \beta))], \tag{7}$$

where the weights $w(y)$ depend on whether the outcome is desired or undesired, and $v_{\mathrm{KTO}}$ adjusts the model based on the reference distribution:

$$v_{\mathrm{KTO}}(x, y; \beta) = \begin{cases} \sigma(r_{\mathrm{KTO}}(x, y) - z_{\mathrm{ref}}) & y \sim y_{\mathrm{desired}}|x \\ \sigma(z_{\mathrm{ref}} - r_{\mathrm{KTO}}(x, y)) & y \sim y_{\mathrm{undesired}}|x \end{cases}, \quad w(y) = \begin{cases} \lambda_U & y \sim y_{\mathrm{undesired}}|x \\ \lambda_D & y \sim y_{\mathrm{desired}}|x \end{cases},$$

where $z_{\mathrm{ref}} = \mathbb{E}_{x' \sim \mathcal{D}}[\beta D_{\mathrm{KL}}(\pi_\theta(y'|x') \| \pi_{\mathrm{ref}}(y'|x')]$ and $r_{\mathrm{KTO}}(x, y) = \beta \log \frac{\pi_\theta(y|x)}{\pi_{\mathrm{ref}}(y|x)}$. Further details of mentioned methods are provided in Appendix A. These methods have been widely adopted in training generative models, including diffusion models like Denoising Diffusion Policy Optimization (DDPO) (Black et al., 2024). This work investigates how these methodologies can be applied to contrastive learning-based models. This paper focuses on two critical applications: mitigating typographic attacks and addressing biases, such as gender bias in contrastive visual models like CLIP.

**Typographic attacks.** Typographic attacks exploit a model's tendency to prioritize text over visual content (Goh et al., 2021), causing CLIP to misclassify images based on misleading text. For example, writing *"cat"* on an image of a dog could lead CLIP to incorrectly label it as *"an image of a cat"*. This represents a clear example of undesired behavior in CLIP and is an ideal test case for preference optimization. Recent works, such as (Azuma & Matsui, 2023; Joanna Materzynska et al., 2022; Ilharco et al., 2022), address this shortcoming through various fine-tuning schemes.

**Inductive biases.** Biases such as gender and racial bias (Ruggeri & Nozza, 2023; Zhang et al., 2022; Bolukbasi et al., 2016; Jialu Wang et al., 2021) are prevalent in vision-language models that often unintentionally encode and maintain societal biases, leading to skewed predictions disproportionately affecting marginalized groups. Another common issue is task bias, where models focus on unintended aspects of the task, further amplifying distortions in their outputs (Menon et al., 2022).

## 3 METHODOLOGY

Figure 1 gives an overview of our approach with a joint training objective: a preference-based contrastive optimization using a dataset of preference labels $\mathcal{D}_{\mathrm{pref}}$ and regularization using a regularization dataset $\mathcal{D}_{\mathrm{reg}}$. Details of our methodology are given in the following.

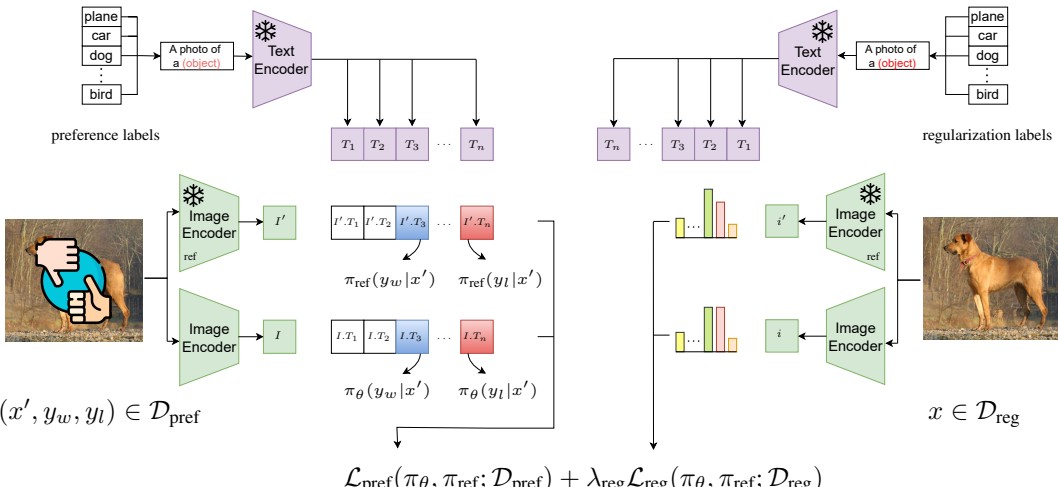

Figure 1: Overview of our proposed approach. On the left side, we calculate the preference optimization loss $\mathcal{L}_{\text{pref}}(\pi_\theta, \pi_{\text{ref}}; \mathcal{D}_{\text{pref}})$ using the preference dataset, and the output logits $\mathcal{T}(y_w)^T\mathcal{I}(x), \mathcal{T}(y_l)^T\mathcal{I}(x)$ from both models. On the right side, the regulatory loss $\mathcal{L}_{\text{reg}}(\pi_\theta, \pi_{\text{ref}}; \mathcal{D}_{\text{reg}})$ is calculated using the regularization dataset. The snowflake icons denote frozen encoders.

## 3.1 PROBLEM FORMULATION: CONTRASTIVE LEARNING AS AN MDP

To effectively apply preference optimization to contrastive learning models, we first need to frame the problem within the reinforcement learning paradigm, modeling it as a Markov decision process (MDP). Specifically, we define the contrastive learning task as a one-step MDP, denoted as $\mathcal{M} = (\mathcal{S}, \mathcal{A}, \rho_0, P, R, \gamma)$, where $\mathcal{S}$ is the state space, $\mathcal{A}$ is the action space, $\rho_0$ is the initial state distribution, $P$ is the transition function, $R$ is the reward function, and $\gamma$ is the discount factor. Since it is a one-step process, the transition function $P$ and the discount factor $\gamma$ are unnecessary, as states do not continue beyond the first step. In this setup, the model operates as a retrieval system: the input (either text or image) represents the state $s \in \mathcal{S}$, and the most similar text or image is selected as the action $a \in \mathcal{A}$. Additionally, $\rho_0$ defines the distribution of initial states, and $R$ represents the reward function.

The policy $\pi_\theta(a|s)$ is defined analogously to the contrastive learning objective, typically using a softmax function over the similarity score $f_\theta(x, y) = \mathcal{I}_\theta(x)^T\mathcal{T}_\theta(y)/\tau$, where $\mathcal{I}_\theta : \mathcal{X}_{\text{image}} \to \mathbb{R}^d$ and $\mathcal{T}_\theta : \mathcal{X}_{\text{text}} \to \mathbb{R}^d$ are the encoders for the image and text, respectively. Here, $\mathcal{X}_{\text{image}}$ and $\mathcal{X}_{\text{text}}$ represent the input distributions of images and texts, and $d$ is the dimensionality of the latent space where both image and text representations are embedded. $\theta$ represents the model parameters, and $\tau$ is the temperature. The action space is chosen from a fixed set of possible outputs $\mathcal{A} = \mathcal{Y} \triangleq \{y_i\}_{i \in \{1, \dots, K\}}$, where $K$ represents the total number of classes. The components of $\mathcal{M}(\mathcal{S}, \mathcal{A}, \rho_0, R)$ are defined as follows:

$$\begin{cases} s \triangleq x \\ a \triangleq y \end{cases} , \quad \begin{cases} \rho_0(s) \triangleq p(x) \\ R(s, a) \triangleq r(x, y) \end{cases} , \quad \pi_\theta(a|s) \triangleq \frac{e^{f_\theta(y,x)}}{\sum_{y_i} e^{f_\theta(y_i,x)}}.$$

This MDP framework can be applied to different tasks and provides flexibility in designing policies. We specifically utilize methods like DPO and KTO due to their implicit reward modeling, which allows for more effective model alignment with desired behaviors.

## 3.2 PREFERENCE AND REGULARIZATION DATASETS

In our framework, the desired response of the model given an input $x$ (e.g., an adversarial image) is considered the preferred output $y_w$. This represents the correct or human-aligned response that we aim to optimize for. On the other hand, the dispreferred output $y_l$ refers to the model's potentially biased response, or the target of an adversarial attack, or an undesirable outcome (such as bias related

to gender or other attributes). The model learns to be robust against adversarial attacks or specific biases by comparing these two outputs.

We use two datasets to facilitate this learning:

- **Preference Dataset** $\mathcal{D}_{\text{pref}}$: This dataset contains pairs $(x, y_w, y_l)$, where $y_w$ is the preferred output and $y_l$ is the dispreferred output. The model learns to differentiate between desired and undesired outcomes through this dataset, improving its robustness against adversarial attacks and biases.
- **Regularization Dataset** $\mathcal{D}_{\text{reg}}$: This dataset consists of clean samples (images or text) without adversarial manipulation. It is used during training to ensure the model retains its accuracy on clean data.

By combining these datasets, the model is trained to align with a specific behavior or to be robust against adversarial inputs while maintaining performance on clean data and other downstream tasks.

## 3.3 PREFERENCE-BASED CONTRASTIVE OPTIMIZATION

Building on the proposed formulation, preference optimization can be effectively applied to contrastive learning models by guiding the optimization process toward desired behaviors. Moreover, we can shape the policy to learn specific behaviors or patterns by defining them as *preferred*. Furthermore, in Section 4.3, we highlight a significant advantage of this optimization paradigm as a fine-tuning method, helping retain the pre-trained knowledge of the model while adapting it to new preferences and support it with experiment results.

Recalling Eq. (4), we first focused on policy ratio, $h_{\pi_\theta}(y_w, y_l, x)$ which we can write as:

$$h_{\pi_\theta}(y_w, y_l, x) = \left( \log \pi_\theta(y_w|x) - \log \pi_\theta(y_l|x) \right) - \left( \log \pi_{\text{ref}}(y_w|x) - \log \pi_{\text{ref}}(y_l|x) \right) \quad (8)$$

Given that the objective of the model $\pi_\theta$ is to simultaneously increase the probability of the preferred label $\pi_\theta(y_w|x)$ and decrease the probability of the dispreferred label $\pi_\theta(y_l|x)$, the term $h_{\pi_\theta}(y_w, y_l, x)$ is positive. Note that $\pi_{\text{ref}}$ in this framework is kept frozen.

**Lemma 3.1** *Under the assumption that the text encoder is frozen, i.e., $\mathcal{T}_{ref} = \mathcal{T}_\theta = \mathcal{T}$, the policy ratio for models using the contrastive learning policy, in the methods such as DPO or IPO can be expressed as:*[2]

$$h_{\pi_\theta}(y_w, y_l, x) = \frac{1}{\tau}(\mathcal{I}_\theta(x) - \mathcal{I}_{ref}(x))^T(\mathcal{T}(y_w) - \mathcal{T}(y_l)). \quad (9)$$

Based on Lemma 3.1, the term $h_{\pi_\theta}$ depends only on the term $(\mathcal{I}_\theta(x) - \mathcal{I}_{\text{ref}}(x))^T(\mathcal{T}(y_w) - \mathcal{T}(y_l))$. This indicates that the loss objective is designed to adjust $\mathcal{I}_\theta(x)$ to align more closely with the preferred text embedding difference, i.e., $\mathcal{T}(y_w) - \mathcal{T}(y_l)$, while maintaining appropriate proximity to the reference embedding $\mathcal{I}_{\text{ref}}(x)$. Having this in mind, we now raise the following question:

### *How does the Gradient update work in case of DPO/IPO?*

First, we need to formulate the gradient of loss correctly in order to analyze it. The following Corollary provides a more insightful way of writing the gradient.

**Corollary 3.2** *Using Lemma 3.1, we can obtain the gradient of the loss as the following.*[3]

$$\nabla_\theta \mathcal{L}_{pref}(\pi_\theta, \pi_{ref}) = - \underbrace{w_{pref}(y_w, y_l; x)}_{(I)} \cdot \underbrace{\left[ \frac{\partial \mathcal{I}_\theta}{\partial \theta} \right]^T (\mathcal{T}(y_w) - \mathcal{T}(y_l))}_{(II)}, \quad (10)$$

where the *gradient weight* (I) in DPO is $w_{\text{pref}}(y_w, y_l; x) = \sigma(-\beta h_{\pi_\theta}(y_w, y_l, x))$, and in IPO, $w_{\text{pref}}(y_w, y_l; x) = \left( \frac{1}{2\beta} - h_{\pi_\theta}(y_w, y_l, x) \right)$. Intuitively, the term (II) shows that adjusting the image embedding $\mathcal{I}_\theta(x)$ in the direction of $\mathcal{T}(y_w) - \mathcal{T}(y_l)$, also encourages the model to better align

---

[2]See Appendix B.1 for the proof.

[3]Further details are provided in Appendix B.3.

with the preferred outputs. Importantly, if the distribution of $\pi_\theta$ deviates significantly from the reference distribution $\pi_{\text{ref}}$, the absolute value of the *policy ratio* increases. As noted previously, this effect reduces the update rate in the direction of (II), acting as a mechanism to control the model and keep it closer to the reference policy, preventing excessive divergence from the reference distribution.

In the KTO method, since there is no direct comparison between $y_w$ and $y_l$, we do not get the same equation as in 9. However, the main idea still applies. The model adjusts the image embedding $\mathcal{I}_\theta(x)$ to match human preferences, but it does so through a different optimization approach.

### 3.3.1 Regularization

In all our training methods, we explicitly constrain the model to remain proximal to the reference model on the preference dataset $\mathcal{D}_{\text{pref}}$. However, the preference dataset $\mathcal{D}_{\text{pref}}$ is typically small and often fails to capture the entire data manifold. In many cases, it primarily consists of adversarial and out-of-distribution samples. To address this limitation, we introduce additional regularization terms into our training process, ensuring the trained model maintains proximity to the reference model. This motivates the introduction of a regularization term $\mathcal{L}_{\text{reg}}$, designed to maintain proximity between the trained model $\pi_\theta$ and the reference model $\pi_{\text{ref}}$ as follow:

$$\mathcal{L}_{\text{reg}}(\pi, \pi_{\text{ref}}; \mathcal{D}_{\text{reg}}) = D_{\text{KL}}(\pi(y|x) \| \pi_{\text{ref}}(y|x)) = \mathbb{E}_{x \sim \mathcal{D}_{\text{reg}}} \mathbb{E}_{y \sim \pi(y|x)} \left[ \log \frac{\pi(y|x)}{\pi_{\text{reg}}(y|x)} \right], \quad (11)$$

where $\mathcal{D}_{\text{reg}}$ is the set of clean images; for example, in the typographic attack task, we construct $\mathcal{D}_{\text{reg}}$ using clean images from the training dataset. By adding this term to the loss function, the model can better maintain its accuracy on true labels, even when facing adversarial challenges, by using the regularization dataset $\mathcal{D}_{\text{reg}}$ to ensure robustness.

A shortcoming of these preference losses also suggests introducing additional regularization terms. From this point on, for the sake of simplicity, we denote $\frac{\pi_\theta(y_w|x)}{\pi_{\text{ref}}(y_w|x)} = x_1$ and $\frac{\pi_\theta(y_l|x)}{\pi_{\text{ref}}(y_l|x)} = x_2$. To better understand the effect of regularization, we will build up on top of the following theorem.

**Theorem 3.3** *(Feng et al., 2024) The partial derivatives of Eq. (5) with respect to $x_1$ and $x_2$ are given by:*

$$\left| \frac{\partial \mathcal{L}_{DPO}(x_1; x_2)}{\partial x_1} \Big/ \frac{\partial \mathcal{L}_{DPO}(x_1; x_2)}{\partial x_2} \right| = \frac{x_2}{x_1} < 1. \quad (12)$$

We find a similar result in the case of IPO:[4]

**Proposition 3.4** *The partial derivatives of Eq. (6) with respect to $x_1$ and $x_2$ are given by:*

$$\left| \frac{\partial \mathcal{L}_{IPO}(x_1; x_2)}{\partial x_1} \Big/ \frac{\partial \mathcal{L}_{IPO}(x_1; x_2)}{\partial x_2} \right| = \frac{x_2}{x_1} < 1. \quad (13)$$

In our problem formulation in Section 3.1, we define the true label as $y_w$ and the adversarial label (e.g., typographic label) as $y_l$. According to Theorem 3.3 and Proposition 3.4, the rate of increase in the probability of the preferred response $y_w$ is lower than the rate of decrease in the probability of the dispreferred response $y_l$, Therefore, as suggested by (Feng et al., 2024), our model primarily focuses on reducing the likelihood of the dispreferred response. This also motivates us to introduce more regulatory terms to ensure proximity.

Finally, for training, we propose using a total loss function from the dataset $\mathcal{D} = (\mathcal{D}_{\text{pref}}, \mathcal{D}_{\text{reg}})$ as follows, where $\mathcal{L}_{\text{reg}}$ is the regularization loss and $\mathcal{L}_{\text{pref}}$ is the preference optimization objective, computed using one of Eqs. 5, 6, or 7:

$$\mathcal{L}(\pi_\theta, \pi_{\text{ref}}; \mathcal{D}) = \mathcal{L}_{\text{pref}}(\pi_\theta, \pi_{\text{ref}}; \mathcal{D}_{\text{pref}}) + \lambda_{\text{reg}} \mathcal{L}_{\text{reg}}(\pi_\theta, \pi_{\text{ref}}; \mathcal{D}_{\text{reg}}). \quad (14)$$

For example, consider giving images as input to CLIP, where the model chooses the corresponding caption from a fixed caption set. The selection process follows the policy we defined, using the softmax function over the similarity scores between the image and each caption in the set, as shown in Figure 1. The training process using the proposed loss (Eq. (14)) is outlined in Algorithm 1.

---

[4]See Appendix B.2 for the proof.

---

**Algorithm 1** Preference-based contrastive optimization

---

**Require:** dataset $\mathcal{D} = (\mathcal{D}_{\text{pref}}, \mathcal{D}_{\text{reg}})$, Model $\pi_\theta$, Reference model $\pi_{\text{ref}}$, Regularization coef. $\lambda_{\text{reg}}$
1: **for** each batch $b \in \mathcal{D}$ **do**
2:     $b_{\text{pref}}, b_{\text{reg}} \leftarrow b$                                                  ▷ Get batch of preference / regularization data
3:     Compute $\pi_\Psi(y|x) \triangleq \text{Softmax}(f_\Psi(y,x))$ for $\Psi \in \{\theta, \text{ref}\}$     ▷ Compute model and ref. distributions
4:     $l_{\text{pref}} \leftarrow \mathcal{L}_{\text{po}}(\pi_\theta, \pi_{\text{ref}}; b_{\text{pref}})$              ▷ Compute preference loss using one of Eqs. (5), (6), or (7)
5:     $l_{\text{reg}} \leftarrow \mathcal{L}_{\text{reg}}(\pi_\theta, \pi_{\text{ref}}; b_{\text{reg}})$              ▷ Compute regularization loss as in Eq. (11), zero if $b_{\text{reg}} = \{\}$
6:     $l_{\text{tot}} \leftarrow l_{\text{pref}} + \lambda_{\text{reg}} \cdot l_{\text{reg}}$                            ▷ Total loss
7:     Update model $\pi_\theta$ by minimizing $l_{\text{tot}}$
8: **end for**

---

## 3.4 LINEAR TRANSFORMATIONS AND ADAPTATIONS

In many cases, fine-tuning the entire model may be unnecessary and require more interpretation and control. Therefore, we employ a linear head on top of the encoders. This linear projection layer offers a more interpretable approach, providing a plug-and-play framework where the projection can be learned during training and then applied as needed.

This linear layer also introduces a linear-algebraic perspective to the adaptation process. Suppose the learned transformation is represented by the learnable matrix $W$. This matrix transforms both vectors $\mathcal{I}(x)$ and $\mathcal{T}(y)$ to $W\mathcal{I}(x)$ and $W\mathcal{T}(y)$ respectively, thereby transforming the similarity function $f(y,x) = \mathcal{I}(x)^T\mathcal{T}(y)/\tau$ into a new function $\tilde{f}(y,x) = \mathcal{I}(x)^T W^T W \mathcal{T}(y)/\tau$. Utilizing the Singular Value Decomposition (SVD), we can express $W = U\Sigma V^T$. Therefore, we have:

$$\tilde{f}(y,x) = \mathcal{I}(x)^T W^T W \mathcal{T}(y)/\tau = (V^T\mathcal{I}(x))^T \Sigma^2 (V^T\mathcal{T}(y))/\tau. \tag{15}$$

In this formulation, the unitary matrix $V$ selects the important directions, and the diagonal matrix $\Sigma = \text{diag}(\sigma_1, \ldots, \sigma_n)$ determines the degree of attenuation or amplification along each direction $v_i$. Specifically, each $\sigma_i$ determines whether the corresponding direction $v_i$ is strengthened ($\sigma_i > 1$) or weakened ($\sigma_i < 1$). As detailed in Appendix E, our training process keeps the model close to the reference, as expected. Consequently, each $\sigma_i$ remains relatively close to 1, and the overall matrix $W^T W$ stays close to the identity matrix.

This observation enables post-training adjustments, allowing us to strengthen or weaken each $v_i$ as needed. One approach to achieve this is by directly tuning each singular value $\sigma_i$, either through linear interpolation or by applying matrix powers. In our work, we utilize matrix powers to modify the singular values via a transformation scaling parameter $t \in \mathbb{R}$, implementing the transformation $\Sigma \to \Sigma^t$ as follows:

$$W_t = U\Sigma^t V^T, \tag{16}$$

where $\Sigma^t = \text{diag}(\sigma_1^t, \ldots, \sigma_n^t)$. This approach provides flexibility in adapting the model, enabling fine-grained control over the transformation applied to the encoded representations.

For large positive or negative values of $t$, we recommend normalizing the output embeddings, as done in the original CLIP model. This normalization prevents certain directions in the embedding space from becoming excessively large. While normalization still allows for the strengthening or weakening of specific directions, it ensures that the length of the vector remains bounded, instead of becoming arbitrarily large. This approach maintains the intended alignment or misalignment with certain directions without causing instability in the embedding space.

## 4 EXPERIMENTS AND RESULTS

In this section, we experimentally justify our method capabilities, using the CLIP model as our contrastive learning model. An explanation of the datasets we used can be found in Appendix F.

**Setup:** For all experiments, we used 8 A100 GPUs, each with 40GB of memory. For the typographic attack experiments, hyperparameters such as $\beta$ and $\lambda$ were selected based on our empirical studies described in the appendix. Specifically, we used $\beta = \lambda = 1$ for DPO, $\beta = \lambda = 0.01$ for IPO, and $\beta = 1.5, \lambda = 0.01$ for KTO. In disentangling gender bias, we also set $\beta = \lambda = 1$. All models were trained for three epochs with a batch size of 512. The learning rate was set to $2 \times 10^{-5}$, and we

Table 1: Classification accuracy on various datasets: O (Original dataset) and T (Typographic dataset). The last row highlights the differences between KTO, our best-performing proposed method, and the best-performing baseline for each dataset. Positive differences indicate improvements achieved by the proposed method.

| Method | Caltech101 | | OxfordPets | | StanfordCars | | Flowers102 | | FGVCAircraft | | DTD | | SUN397 | | EuroSAT | | Avg. | |
|---|---|---|---|---|---|---|---|---|---|---|---|---|---|---|---|---|---|---|
| | O | T | O | T | O | T | O | T | O | T | O | T | O | T | O | T | O | T |
| CLIP | 88.64 | 63.97 | 87.35 | 58.95 | 58.72 | 21.02 | 66.32 | 31.32 | 18.99 | 10.83 | 44.57 | 25.53 | 61.74 | 34.02 | 42.98 | 4.86 | 58.66 | 31.31 |
| Materzynska+ | 80.53 | 74.73 | 75.01 | 63.61 | 40.33 | 15.79 | 51.86 | 34.95 | 13.23 | 8.28 | 36.28 | 33.03 | 51.06 | 39.52 | 37.32 | 16.22 | 48.25 | 35.77 |
| PAINT | 88.48 | 83.57 | 85.23 | 76.53 | 55.30 | 33.44 | **64.73** | 54.92 | 17.73 | 14.46 | **42.61** | 36.60 | 61.69 | 53.62 | 38.20 | 17.31 | 56.74 | 46.31 |
| Defense-Prefix | **89.28** | 79.54 | **87.22** | 72.86 | 57.47 | 28.64 | 63.82 | 44.12 | **19.26** | 14.49 | 40.64 | 31.60 | 61.41 | 43.50 | 43.85 | 9.85 | **57.87** | 40.58 |
| Ours (DPO) | 87.50 | 85.43 | 85.25 | 79.72 | 56.03 | 34.33 | 56.60 | 55.70 | 16.21 | 13.87 | 39.36 | 38.48 | 61.02 | 56.34 | **49.33** | 28.32 | 56.41 | 49.02 |
| Ours (IPO) | 85.73 | 83.78 | 85.32 | 80.44 | 53.67 | 35.02 | 54.50 | 52.80 | 17.97 | **15.86** | 40.53 | 39.94 | **61.91** | 58.05 | 46.12 | **43.23** | 55.72 | 51.14 |
| Ours (KTO) | 87.67 | **86.02** | 85.41 | **81.02** | **57.76** | **37.04** | 59.10 | **58.00** | 17.27 | 15.59 | 40.74 | **40.33** | 62.52 | **59.01** | 46.26 | 36.94 | 57.09 | **51.74** |
| Difference | ↓1.61 | ↑2.45 | ↓1.81 | ↑4.47 | ↑0.9 | ↑3.60 | ↓5.63 | ↑3.08 | ↓1.99 | ↑1.10 | ↓1.87 | ↑3.73 | ↑0.83 | ↑5.39 | ↑2.41 | ↑19.63 | ↓0.78 | ↑5.43 |

employed the Adamax optimizer with a linear warmup and a cosine scheduler, setting the warmup ratio to 0.1. We set the random seed to 0 for all experiments to ensure reproducibility.

## 4.1 TYPOGRAPHIC ROBUSTNESS

We further evaluate our method on the typographic attack mitigation task. We set up the problem as described in Section 3, using the template `"an image of a <class i>"` (or a more dataset-specific template for non-generic images), designating the typographic class as the undesired outcome $y_l$ and the original caption as the desired outcome $y_w$.

**Baselines:** We evaluate our method against the following baselines: pre-trained CLIP (Radford et al., 2021), (Materzynska et al., 2022), PAINT (Ilharco et al., 2022), and Defense-prefix (Azuma & Matsui, 2023), with results collected from (Azuma & Matsui, 2023) for comparability. (Materzynska et al., 2022) propose using orthogonal projections to create subspaces in CLIP's image encoder that separate the processing of written words from visual concepts. PAINT (Patching with Interpolation) is a technique for adapting open-vocabulary models to downstream tasks, while Defense-Prefix is specifically designed to counter typographic attacks on CLIP by inserting a learned token before class names in text prompts. We use ImageNet-100, a 100-class subset of ImageNet (Russakovsky et al., 2015), to train the model. Typographic attack images are generated by adding misleading text labels to the original images using open-source implementations from (Azuma & Matsui, 2023). Further details of our baselines are provided in Appendix F.

Our results shown in Table 1 indicate that our methods effectively prevent typographic attacks with little loss of prior knowledge, with improvements of up to 19.63 in accuracy over the best performing baseline.[5]

### 4.1.1 CONTROL BETWEEN OPTICAL CHARACTER RECOGNITION AND OBJECT DETECTION

In this experiment, we apply linear adaptations and retrain the network, then evaluate the training results. We then interpolate between models using previously discussed linear transformation methods, varying the transformation scaling $t$, defined in Eq. (16) from $-4.0$ to $4.0$ to assess performance. By doing this, we measure the effectiveness of our controlling scheme between the tasks of Optical Character Recognition (OCR) and Object Detection (OD).

Our results in Figure 2 demonstrate that we can confidently control the trade-off between OCR and OD without any additional training and with minimal impact on overall accuracy. This method allows us to choose which task to prioritize flexibly, and we can visualize the performance frontier, showing that improvements in OCR accuracy come at the cost of OD accuracy and vice versa. The ability of this linear layer to manage the trade-off suggests that the concept is linearly separable within the CLIP embedding space.

---

[5]More analyses are shown in Appendices C, D and J.

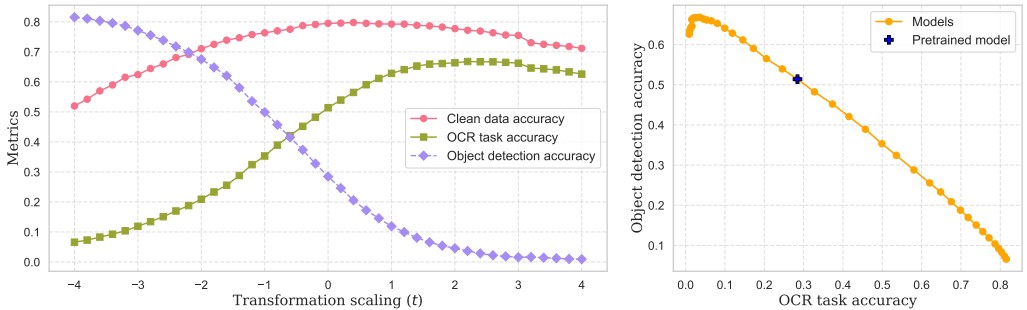

(a) Accuracy on typographic samples and percentage of typographic label predictions versus transformation scaling factor $t$. As $t$ increases, the model favors object labels over typographic labels while maintaining accuracy.

(b) Frontier of a DPO fine-tuned model, showing OCR vs. OD accuracy across with varying $t$.

Figure 2: Comparisons of optical character recognition (OCR) and object detection (OD).

## 4.2 DISENTANGLING GENDER UNDERSTANDING

In this setup, we demonstrate how our approach can reverse a contrastive learning model's understanding of a concept (e.g., gender) without significantly altering other aspects of its performance.

For demonstration, we focus on flipping CLIP's understanding of gender. We utilize a dataset of labeled images featuring men and women engaged in various activities and occupations from (Zhang et al., 2022). The problem setup is as follows: given choices like $y_{\text{man}}$ = "The man is <activity>" and $y_{\text{woman}}$ = "The woman is <activity>", the CLIP model typically picks one of these answers. To reverse the gender understanding, we designate the label with the correct activity but flipped gender as the preferred answer $y_w$ and the original gender-specific label as $y_l$. As the regulatory term, we use the same dataset of images. However, we employ androgynous target labels such "The person in <activity>" to ensure that our model does not forget any information on detecting the correct activity. We repeat our learning process using only the linear layer and then visualize the effect of varying the transformation scaling $t$ between 0 and 1. Our results confirm that we have a high degree of control over CLIP's understanding of gender, as demonstrated in Figure 3.

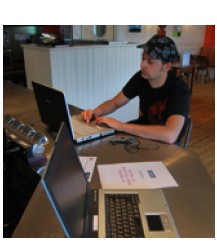
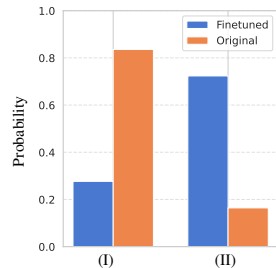
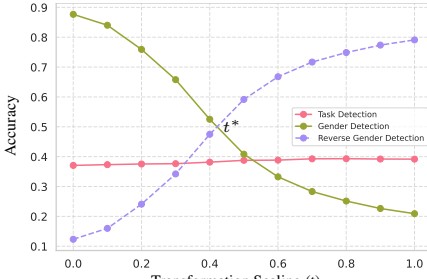

(a) Example image showing a man working.

(b) Model predictions before and after applying our gender-flipping method, showing changes in the predicted captions:
(I) "The man in the photo is working."
(II) "The woman in the photo is working."

(c) As $t$ increases from 0 to 1, gender-specific predictions are reversed. $t^*$ marks the point where gender information is neutralized, leading to balanced male and female predictions. Task detection accuracy remains stable across all models.

Figure 3: Analyses of the models' understanding of gender.

To further evaluate our results, we apply the modified CLIP models to an image retrieval task, distinct from the original text retrieval task used for training. We assess how this impacts the retrieved images for specific text prompts by computing the similarity scores between the prompts and images as $s(\text{image}, \text{text}) = \mathcal{I}_\theta(\text{image})^T \mathcal{T}_\theta(\text{text})$. Using the dataset from (Zhang et al., 2022), we issue

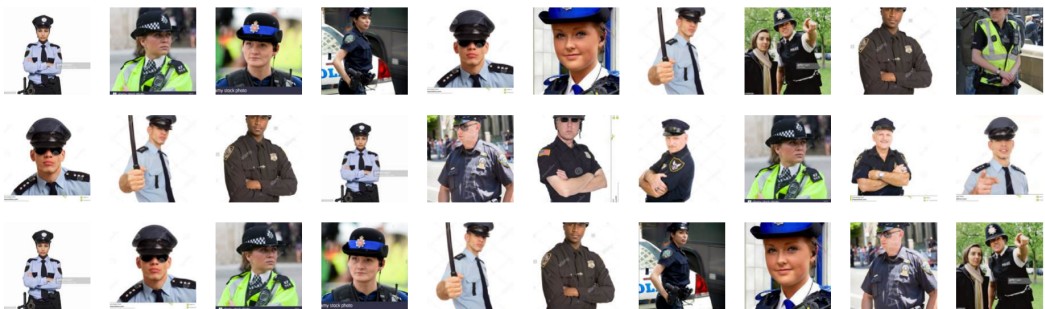

Figure 4: Images retrieved for the caption *"an image of a police"* with three different policies from top to bottom: reversed understanding of gender (6W, 4M), pretrained CLIP model (2W, 8M), neutralized understanding of gender (5W, 5M), i.e., $t = t^*$.

prompts describing individuals in various occupations to analyze the model's behavior and showcase an instance in Figure 4.

### 4.3 PRESERVING PRETRAINED KNOWLEDGE VIA PREFERENCE OPTIMIZATION

In all of the aforementioned preference optimization methods, an implicit or explicit constraint exists to keep the model as close as possible to the reference model. Specifically, including KL-divergence regularization facilitates adjustments without deviating significantly from the reference model $\pi_{\text{ref}}$. To support this claim, Figure 5a shows the KL-divergence between the output distributions of the model and the reference model on the original dataset, measured as $\mathbb{E}_{x \sim \mathcal{D}_{\text{reg}}} [D_{\text{KL}}(\pi_\theta(\cdot|x)||\pi_{\text{ref}}(\cdot|x))]$, after fine-tuning for the typographic robustness task using different optimization methods.

Compared to Cross-Entropy optimization, our proposed methods result in lower divergence, demonstrating better retention of the model's pre-trained knowledge. Figure 5b illustrates that we can control this deviation (i.e., retention of pre-trained knowledge) during fine-tuning by adjusting the $\beta$ hyperparameter, consistent with Eq. (3). All models in (a) and (b) were trained on the *Imagenet100* and *FOOD101* datasets, respectively, and the models in (b) were optimized using the IPO method.

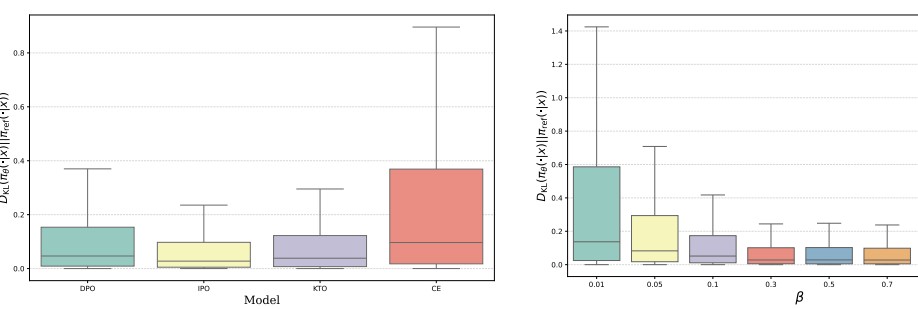

(a) Our methods vs. Cross-Entropy optimization (CE).   (b) Effect of changing the hyperparameter $\beta$.

Figure 5: KL-divergence studies.

## 5 CONCLUSION

This paper introduces a novel approach for improving contrastive learning models through Preference Optimization (PO). Our approach systematically aligns model behavior with desired preferences, enhancing robustness against adversarial attacks and effectively mitigating biases in vision-language models like CLIP. Our experiments demonstrate improved performance over standard contrastive learning methods in adversarial robustness and fairness. This work could open new pathways for applying preference-based optimization to non-generative models.

ACKNOWLEDGMENTS

This work was partially funded by the Federal Ministry for Digital and Transport (BMDV), Germany ("MoToRes", 19F2271A) and by the Federal Ministry for Economic Affairs and Climate Action (BMWK), Germany ("ATTENTION!", 01MJ22012D).

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

# Appendix

## Table of Contents

## A    THEORETICAL FOUNDATIONS OF RLHF

RLHF extends the traditional reinforcement learning paradigm by incorporating human preferences into the learning process (Ouyang et al., 2022). This feedback can take the form of direct preference comparisons between different actions or sequences of actions, or it can involve explicit corrections when the agent's behavior is undesirable. Human judgments are converted into a reward signal that helps the agent refine its policy. RLHF has gained attention in areas like natural language processing and fine-tuning large language models, where specifying a clear reward function is difficult and aligning the agent's behavior with human values or intentions is critical. This method offers a more robust approach to training AI systems in tasks that require subjective judgment or nuanced decision-making.

In the context of aligning Large Language Models (LLMs), RLHF plays a crucial role. The process usually begins with supervised fine-tuning (SFT) of a pre-trained model on task-specific datasets, which helps the model learn to generate appropriate responses for various queries, resulting in a reference model denoted as $\pi_{\text{ref}}$. This model is refined using RLHF, which consists of three core steps.

*1. Supervised Fine-tuning (SFT):* Pre-trained language models often require an initial supervised fine-tuning phase to follow human instructions better. This phase aligns the model's responses with human expectations. After this phase, we obtain a model $\pi_\theta^{\text{SFT}}$, which serves as the initial policy for further training.

*2. Reward Model Training:* Once the SFT model is trained, the next step in RLHF is to train a reward model $r_\psi(x, y)$, where $x$ is an input prompt and $y$ is the corresponding response. The reward model is trained to reflect human preferences between different responses. Given two responses $y_w$ (preferred, or "win") and $y_l$ (less preferred, or "lose"), the Bradley-Terry model is commonly used to model human preferences:

$$p^*(y_w \succ y_l | x) = \frac{e^{r_\psi(x, y_w)}}{e^{r_\psi(x, y_w)} + e^{r_\psi(x, y_l)}}, \tag{17}$$

which can be rewritten as:

$$p^*(y_w \succ y_l | x) = \sigma(r_\psi(x, y_w) - r_\psi(x, y_l)), \tag{18}$$

where $\sigma$ is the sigmoid function. To train the reward model, the following log-loss function is minimized over a dataset of human preferences $\mathcal{D} = \{(x^{(i)}, (y_w^{(i)}, y_l^{(i)}))\}_{i=1}^N$:

$$-\mathbb{E}_{(x, y_w, y_l) \sim \mathcal{D}}[\log \sigma(r_\psi(x, y_w) - r_\psi(x, y_l))]. \tag{19}$$

*3. Fine-tuning with Reinforcement Learning:* In the final stage of RLHF, the trained reward model is kept fixed, and the policy $\pi_\theta^{\text{RL}}$ (initialized from $\pi_\theta^{\text{SFT}}$) is fine-tuned using the PPO algorithm. The goal is to optimize the policy to maximize the expected reward predicted by the reward model while maintaining closeness to the supervised fine-tuned model through Kullback-Leibler (KL) divergence regularization. The optimization problem is formalized as:

$$\max_\theta \mathbb{E}_{x \sim \mathcal{D}, y \sim \pi_\theta^{\text{RL}}(y|x)}[r_\psi(x, y) - \beta\, D_{\text{KL}}(\pi_\theta^{\text{RL}}(y|x) \| \pi_\theta^{\text{SFT}}(y|x))], \tag{20}$$

where $D_{\text{KL}}(\pi_\theta^{\text{RL}}(y|x) \| \pi_\theta^{\text{SFT}}(y|x))$ represents the KL divergence between the RL policy and the SFT policy. The coefficient $\beta$ controls the trade-off between encouraging exploration and maintaining consistency with the supervised model. KL divergence serves two purposes: it acts as an entropy bonus, encourages exploration, and ensures that the RL policy stays within the supervised fine-tuned model.

This structured process ensures that LLMs are capable of solving complex tasks and remain aligned with human preferences and ethical guidelines, which is essential in applications where human values play a crucial role.

### A.1    DERIVING PREFERENCE OPTIMIZATION AS A SUPERVISED LEARNING FRAMEWORK

In traditional RLHF, RL algorithms like PPO are required to optimize the policy since the reward signal is not differentiable. However, RLHF is often slow, primarily due to the need for sampling

generations, and could be more stable in practice, especially in distributed settings. As a result, recent research has focused on designing closed-form loss functions that maximize the margin between preferred and dispreferred generations. One prominent approach is Direct Preference Optimization, which has gained popularity due to its mathematical equivalence with RLHF.

**Direct Preference Optimization (DPO).** DPO (Rafailov et al., 2024) leverages offline preference data to directly optimize the policy without relying on reinforcement learning-based methods like PPO. DPO demonstrates that the optimal solution to Eq. (20), $\pi_\theta^*$, satisfies the following equation:

$$r_\theta(x, y) = \beta \log \frac{\pi_\theta(y|x)}{\pi_{\text{ref}}(y|x)} + \beta \log Z(x). \tag{21}$$

Here, $r_\theta$ represents the reward model, $\pi_\theta$ is the policy model, and $\pi_{\text{ref}}$ is the reference model. Both $\pi_\theta$ and $\pi_{\text{ref}}$ are initialized from the same SFT (Supervised Fine-Tuning) model, but only $\pi_\theta$ is further optimized during DPO, while $\pi_{\text{ref}}$ remains unchanged. $Z(x)$ is the partition function, and $\beta$ is a hyper-parameter that controls the intensity of the reward signal.

The preference probability $P_\theta$ is derived from pairwise comparisons using the Bradley-Terry model. Substituting Eq. (21) into Eq. (19) yields the following loss function for DPO:

$$\mathcal{L}_{\text{dpo}}(\pi_\theta; \pi_{\text{ref}}) = -\mathbb{E}_{(x, y_w, y_l) \sim \mathcal{D}} \left[ \log \sigma \left( \beta \log \frac{\pi_\theta(y_w|x)}{\pi_{\text{ref}}(y_w|x)} - \beta \log \frac{\pi_\theta(y_l|x)}{\pi_{\text{ref}}(y_l|x)} \right) \right]. \tag{22}$$

In Eq. (22), $\sigma$ denotes the sigmoid function, and $\mathcal{D}$ represents the dataset consisting of pairwise preferences. Each preference triplet $(x, y_w, y_l)$ includes a prompt $x$, a preferred response $y_w$, and a less preferred response $y_l$. This formulation allows DPO to optimize directly on pairwise preference data, avoiding the instability of RL methods while maintaining mathematical equivalence with traditional RLHF.

**Identity Preference Optimization (IPO).** As the original DPO loss function has shown limitations in practice, such as overfitting to the preference dataset (Gheshlaghi Azar et al., 2024), IPO was proposed as an extension. IPO introduces a regularization term to the DPO loss to mitigate overfitting by controlling the gap between the log-likelihood ratios of the preferred and dispreferred outputs for both the model and the reference model. The IPO loss function is defined as:

$$\mathcal{L}_{\text{IPO}}(\pi_\theta; \pi_{\text{ref}}) = -\mathbb{E}_{(x, y_w, y_l) \sim \mathcal{D}} \left[ \left( \log \left( \frac{\pi_\theta(y_w|x)\pi_{\text{ref}}(y_l|x)}{\pi_\theta(y_l|x)\pi_{\text{ref}}(y_w|x)} \right) - \frac{\beta^{-1}}{2} \right)^2 \right]. \tag{23}$$

By adding this regularization, IPO ensures better generalization of the model, avoiding overfitting to specific preference patterns and maintaining stability in performance across different datasets.

**Kahneman-Tversky Optimization (KTO).** In language modeling, collecting large-scale preference datasets can be difficult, but it is relatively easier to determine whether an output is desired or undesired. This challenge has inspired KTO (Ethayarajh et al., 2024), a method that bypasses the need for detailed preference data by using a binary signal to guide optimization. KTO is based on prospect theory (Tversky, 2016), directly maximizing the utility of generations using a model of human decision-making.

KTO has demonstrated higher data efficiency and better handling of imbalanced datasets compared to DPO. The KTO loss function is defined as:

$$\mathcal{L}_{\text{KTO}}(\pi_\theta, \pi_{\text{ref}}) = \mathbb{E}_{x, y \sim \mathcal{D}}[w(y)(1 - v_{\text{KTO}}(x, y; \beta))], \tag{24}$$

where $v_{\text{KTO}}(x, y; \beta)$ adjusts based on whether the output is desired or not:

$$v_{\text{KTO}}(x, y; \beta) = \begin{cases} \sigma(r_{\text{KTO}}(x, y) - z_{\text{ref}}) & y \sim y_{\text{desired}}|x \\ \sigma(z_{\text{ref}} - r_{\text{KTO}}(x, y)) & y \sim y_{\text{undesired}}|x \end{cases}, \quad w(y) = \begin{cases} \lambda_U & y \sim y_{\text{undesired}}|x \\ \lambda_D & y \sim y_{\text{desired}}|x \end{cases}.$$

Here, $z_{\text{ref}}$ represents the reference reward, and $r_{\text{KTO}}(x, y) = \beta \log \frac{\pi_\theta(y|x)}{\pi_{\text{ref}}(y|x)}$. The weighting function $w(y)$ differentiates between desired and undesired outputs, where $\lambda_D$ and $\lambda_U$ are hyper-parameters controlling the weighting of each.

KTO's key advantage is its ability to focus on maximizing the reward for desired outputs without inflating the KL divergence term, effectively balancing utility and regularization. This makes KTO a practical solution for real-world applications, where collecting detailed preference data is infeasible.

In all three methods—DPO, IPO, and KTO—the policy $\pi_\theta$ is directly optimized without the need for a reward model. We incorporate these methods into our experiments to compare their effectiveness in various aspects, such as reducing bias and improving adversarial robustness and overall model performance.

## B  PROOFS

For simplicity, in all the sections that follow, we assume $\tau = 1$.

### B.1  PROOF OF LEMMA 3.1

As seen in DPO and IPO,for preference optimization, using a preference dataset he key objective is to optimize based on a preference dataset by calculating $h_\pi(y_w, y_l, x)$ define in 4. using our defined policy $\pi_\theta$ with the softmax output, we find the following;

$$
h_{\pi_\theta}(y_w, y_l, x) = \log\left(\frac{\pi_\theta(y_w|x)\pi_{\text{ref}}(y_l|x)}{\pi_\theta(y_l|x)\pi_{\text{ref}}(y_w|x)}\right) = \log\left(\frac{\frac{e^{f_\theta(y_w,x)}}{\sum_{y_i} e^{f_\theta(y_i,x)}} \cdot \frac{e^{f_{\text{ref}}(y_l,x)}}{\sum_{y_i} e^{f_{\text{ref}}(y_i,x)}}}{\frac{e^{f_\theta(y_l,x)}}{\sum_{y_i} e^{f_\theta(y_i,x)}} \cdot \frac{e^{f_{\text{ref}}(y_w,x)}}{\sum_{y_i} e^{f_{\text{ref}}(y_i,x)}}}\right)
$$

$$
= \log\left(e^{f_\theta(y_w,x)-f_\theta(y_l,x)+f_{\text{ref}}(y_l,x)-f_{\text{ref}}(y_w,x)}\right)
$$

$$
= f_\theta(y_w,x) - f_\theta(y_l,x) + f_{\text{ref}}(y_l,x) - f_{\text{ref}}(y_w,x)
$$

$$
= \left(\mathcal{I}_\theta(x)^T\mathcal{T}_\theta(y_w) - \mathcal{I}_\theta(x)^T\mathcal{T}_\theta(y_l) - \mathcal{I}_{\text{ref}}(x)^T\mathcal{T}_{\text{ref}}(y_w) + \mathcal{I}_{\text{ref}}(x)^T\mathcal{T}_{\text{ref}}(y_w)\right)/\tau
$$

$$
= \mathcal{I}_\theta(x)^T(\mathcal{T}_\theta(y_w) - \mathcal{T}_\theta(y_l)) - \mathcal{I}_{\text{ref}}(x)^T(\mathcal{T}_{\text{ref}}(y_w) - \mathcal{T}_{\text{ref}}(y_w)). \tag{25}
$$

Assuming the text encoder $\mathcal{T}_\theta$ is frozen, such that $\mathcal{T}_\theta = \mathcal{T}_{\text{ref}} = \mathcal{T}$, the expression simplifies to:

$$
h_{\pi_\theta}(y_w, y_l, x) = (\mathcal{I}_\theta(x) - \mathcal{I}_{\text{ref}}(x))^T(\mathcal{T}(y_w) - \mathcal{T}(y_l)). \tag{26}
$$

Our loss functions aim to increase $h_{\pi_\theta}(y_w, y_l, x)$, starting from the initialized value of 0 (since $\pi_\theta$ is initially set to $\pi_{\text{ref}}$). With some proximity constraints, as in DPO and IPO:

$$
\mathcal{L}_{\text{DPO}}(\pi_\theta, \pi_{ref}) = \mathbb{E}_{(x,y_l,y_w)\sim\mathcal{D}}\left[-\log\sigma\left(\beta h_{\pi_\theta}(y_w, y_l, x)\right)\right], \tag{27}
$$

$$
\mathcal{L}_{\text{IPO}}(\pi_\theta; \pi_{\text{ref}}) = \mathbb{E}_{(x,y_w,y_l)\sim\mathcal{D}}\left[\left(h_{\pi_\theta}(y_w, y_l, x) - \frac{\beta^{-1}}{2}\right)^2\right]. \tag{28}
$$

### B.2  PROOF OF PROPOSITION 3.4

By rewriting $\mathcal{L}_{\text{IPO}}(\pi_\theta, \pi_{\text{ref}})$ with $x_1$ and $x_2$, we obtain:

$$
\mathcal{L}_{\text{IPO}}(\pi_\theta, \pi_{\text{ref}}) = \mathbb{E}_{(x_1,y_1,y_2)\sim\mathcal{D}}\left[\left(h_{\pi_\theta}(y_1, y_2, x_1, x_2) - \frac{\beta^{-1}}{2}\right)^2\right]
$$

$$
= \mathbb{E}_{(x_1,y_1,y_2)\sim\mathcal{D}}\left[\left(\log\left(\frac{x_1}{x_2}\right) - \frac{\beta^{-1}}{2}\right)^2\right]. \tag{29}
$$

For $\frac{\partial\mathcal{L}_{\text{IPO}}(x_1;x_2)}{\partial x_1}$,

$$
\frac{\partial\mathcal{L}_{\text{IPO}}(x_1;x_2)}{\partial x_1} = 2\left(\log\left(\frac{x_1}{x_2}\right) - \frac{\beta^{-1}}{2}\right)\cdot\frac{1}{x_2}\frac{x_2}{x_1}
$$

$$
= 2\left(\log\left(\frac{x_1}{x_2}\right) - \frac{\beta^{-1}}{2}\right)\cdot\frac{1}{x_1}. \tag{30}
$$

For $\frac{\partial \mathcal{L}_{\text{IPO}}(x_1;x_2)}{\partial x_2}$,

$$\frac{\partial \mathcal{L}_{\text{IPO}}(x_1;x_2)}{\partial x_2} = 2 \left( \log \left( \frac{x_1}{x_2} \right) - \frac{\beta^{-1}}{2} \right) \cdot \left( -\frac{x_1}{x_2^2} \frac{x_2}{x_1} \right)$$
$$= 2 \left( \log \left( \frac{x_1}{x_2} \right) - \frac{\beta^{-1}}{2} \right) \cdot \frac{-1}{x_2}$$

(31)

and for the second part we have:

$$\left| \frac{\partial \mathcal{L}_{\text{DPO}}(x_1;x_2)}{\partial x_1} \Big/ \frac{\partial \mathcal{L}_{\text{DPO}}(x_1;x_2)}{\partial x_2} \right| = \frac{2 \left( \log \left( \frac{x_1}{x_2} \right) - \frac{\beta^{-1}}{2} \right) \cdot \frac{1}{x_1}}{2 \left( \log \left( \frac{x_1}{x_2} \right) - \frac{\beta^{-1}}{2} \right) \cdot \frac{1}{x_2}},$$
$$= \frac{x_2}{x_1},$$

(32)

and finally given that $x_1 = \frac{\pi_\theta(y_w|x)}{\pi_{\text{ref}}(y_w|x)}$ and $x_2 = \frac{\pi_\theta(y_l|x)}{\pi_{\text{ref}}(y_l|x)}$ are two probability ratios, where $\pi_\theta(y|x) \in [0,1]$ and $\pi_{\text{ref}}(y|x) \in [0,1]$. Assuming $\pi_{\text{ref}}(y|x)$ is the probability of the fixed reference model, we can assume $\pi_{\text{ref}}(y_w|x) = \frac{1}{a}$ and $\pi_{\text{ref}}(y_l|x) = \frac{1}{b}$, where $(a,b \geq 1)$. In this case, we have $x_1 \in [0,a]$ and $x_2 \in [0,b]$. As the DPO optimization progresses, $x_1$ tends to increase and $x_2$ tends to decrease. Consequently, $\pi_\theta(y_w|x)$ will be greater than $\frac{1}{a}$, and $\pi_\theta(y_l|x)$ will be smaller than $\frac{1}{b}$. In other words, this implies that $x_1 = \frac{\pi_\theta(y_w|x)}{\pi_{\text{ref}}(y_w|x)}$ is greater than 1, $x_2 = \frac{\pi_\theta(y_l|x)}{\pi_{\text{ref}}(y_l|x)}$ is less than 1, and therefore $x_2 < x_1$.

### B.3 THE GRADIENT OF THE DPO AND IPO OBJECTIVES

The gradients of these loss functions highlight how the image encoder $\mathcal{I}_\theta$ is adjusted to align more closely with the desired output:

$$\nabla_\theta \mathcal{L}_{\text{DPO}}(\pi_\theta, \pi_{ref}) = -\beta \bigg[ \underbrace{\sigma \Big( -\beta h_{\pi_\theta}(y_w, y_l, x) \Big)}_{\text{higher when proximal to reference}} \cdot \underbrace{\left[ \frac{\partial \mathcal{I}_\theta}{\partial \theta} \right]^T (\mathcal{T}(y_w) - \mathcal{T}(y_l))}_{\text{increase } I_\theta(x) \text{ in direction of } \mathcal{T}(y_w) - \mathcal{T}(y_l)} \bigg]. \quad (33)$$

$$\nabla_\theta \mathcal{L}_{\text{IPO}}(\pi_\theta, \pi_{ref}) = -\bigg[ \underbrace{\left( \frac{\beta^{-1}}{2} - h_{\pi_\theta}(y_w, y_l, x) \right)}_{\text{positive when proximal to reference}} \cdot \underbrace{\left[ \frac{\partial \mathcal{I}_\theta}{\partial \theta} \right]^T (\mathcal{T}(y_w) - \mathcal{T}(y_l))}_{\text{increase } I_\theta(x) \text{ in direction of } \mathcal{T}(y_w) - \mathcal{T}(y_l)} \bigg]. \quad (34)$$

These gradients show that both DPO and IPO adjust the image embedding $\mathcal{I}_\theta(x)$ in the direction of $\mathcal{T}(y_w) - \mathcal{T}(y_l)$, encouraging the model to better align with the preferred outputs.

## C FEATURES LEARNED BY OUR METHOD

In Figure 6, we show saliency maps of vanilla CLIP and our fine-tuned model on images with typographic attacks. CLIP focuses on irrelevant parts of images, especially typographic texts, which results in false predictions. On the other hand, our model is capable of ignoring the text and focuses on the visual concept of the image, which results in a correct prediction.

Further, in Figure 7, we provide the predictions of several models on another example image with a typographic attack. While the fine-tuned model remains robust, correctly classifying the attacked image, the original model misclassifies it. The inverse model confidently selects the typographic label but preserves classification accuracy.

These analyses showcase how applying our method forces the model to focus on the suitable features of the data rather than different sorts of attacks such as typographic.

## D LATENT VISUALIZATION USING VQGAN

In this section, we further investigate our fine-tuned CLIP on typographical attack datasets. We use VQGAN-CLIP (Crowson et al., 2022) as the generative model which uses CLIP in its backbone.

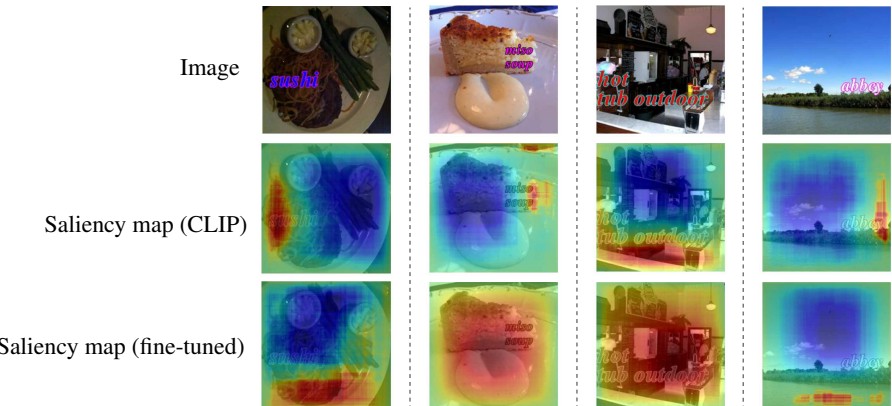

Image

Saliency map (CLIP)

Saliency map (fine-tuned)

Figure 6: Saliency maps of vanilla CLIP and our fine-tuned model given four different images with typographic attacks.

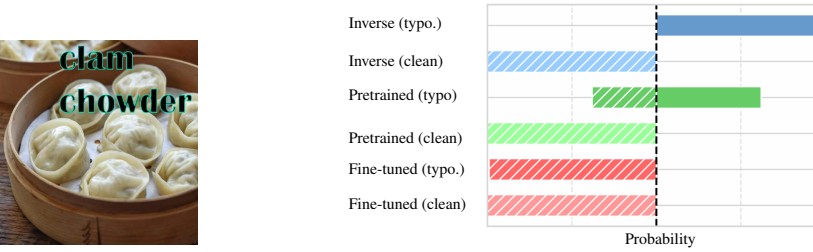

(a) Example image showing dumplings after a typographic attack with misleading text ("clam chowder").

(b) Predictions of six models.

Figure 7: Model predictions given an example image with a typographic attack.

The results are shown in Figure 8. As can be seen, our fine-tuned CLIP completely ignores text in its image encoder representation and only focuses on visual concepts which increase its reliability and trustworthiness.

## E    NEAR ORTHOGONALITY OF $W$

Previous works such as (Materzynska et al., 2022) also utilized a linear projection for their optimization objective. However, when these projections stray away from near orthogonal projections, our pretrained knowledge might be lost; this is a simple consequence of the following fact:

$$\left. \begin{array}{l} \tilde{f}(y,x) = \mathcal{I}(x)^\top W^\top W \mathcal{T}(y) \\ f(y,x) = \mathcal{I}(x)^\top \mathcal{T}(y) \end{array} \right\} \Rightarrow \tilde{f}(y,x) - f(y,x) = \mathcal{I}(x)^\top (W^\top W - I)\mathcal{T}(y)$$

$$\leq \|\mathcal{I}(x)\|\|\mathcal{T}(y)\|\|W^\top W - I\| \leq \|W^\top W - I\|, \quad (35)$$

where the second equality is a result of CLIPs' normalized embeddings, as it is evident from Eq. (35), our deviation from the base model is controlled by $\|W^\top W - I\|$, and ideally, we do not want the deviation to be too large. Previous works such as (Materzynska et al., 2022) added explicit loss objectives such as $R(W) = \|W^\top W - I\|^F$ to maintain proximity, and they have demonstrated the

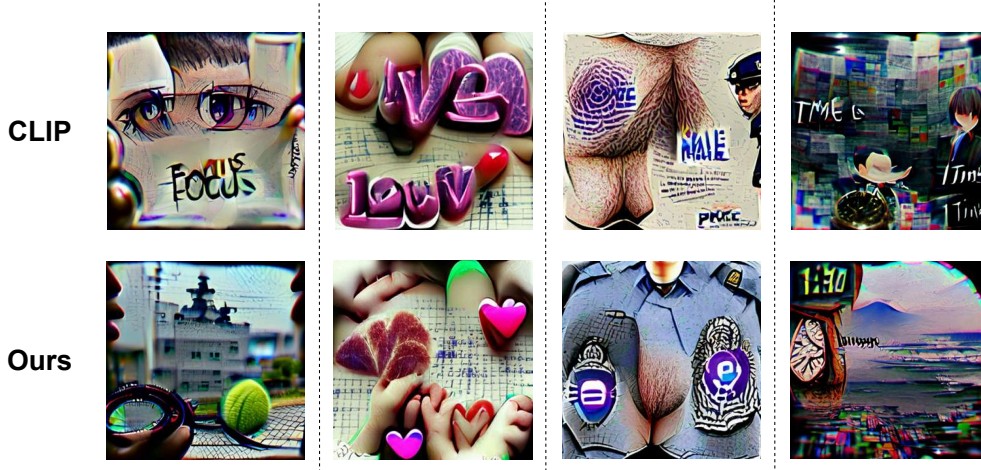

Figure 8: Retrieved images using VQGAN-CLIP (Crowson et al., 2022) using the captions *"focus"*, *"Love"*, *"Male police"* and *"Time"* for image generation. As can be seen, vanilla CLIP puts the texts on the images, but our model fine-tuned on the typographical attack dataset SUN ignores such image texts and generates only visual concepts.

importance of this orthogonality loss. However, our method does not let $\pi$ and $\pi_{\text{ref}}$ diverge easily. Therefore, we expect unitary by default. After training the model using our method, considering that pretrained knowledge has been maintained, we already expect the matrix $W$ to be almost orthogonal. Nonetheless, we quantify this proximity in Figure 9.

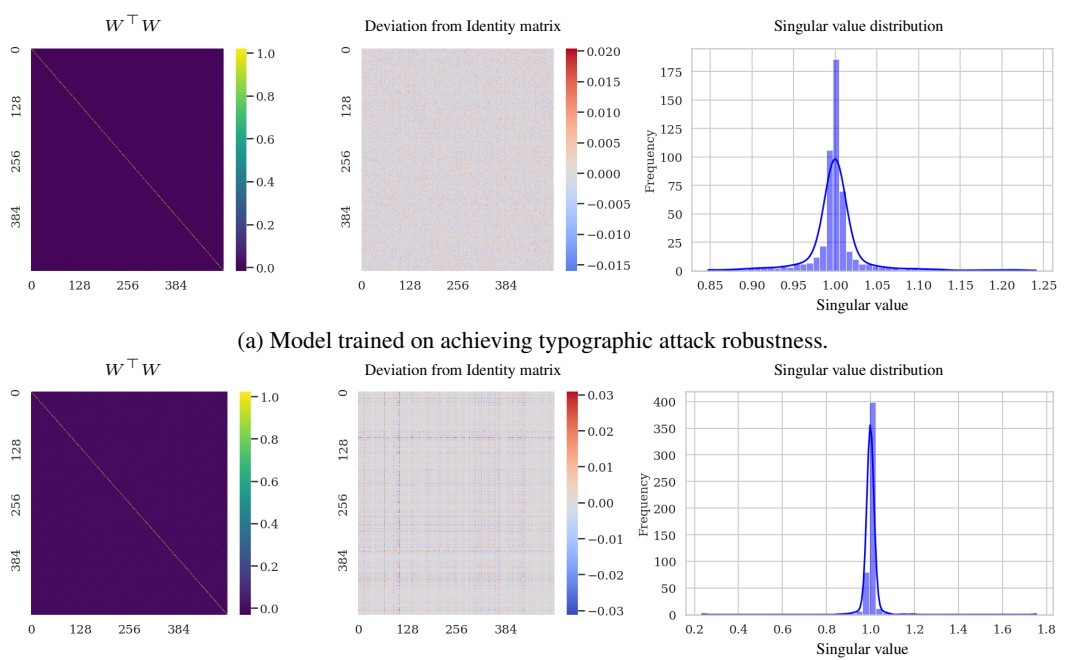

(a) Model trained on achieving typographic attack robustness.

(b) Model trained on reversing gender understanding.

Figure 9: On the orthogonality of $W$: The left plots show the overall heatmaps of $W^\top W$, where it is evident that the matrix is close to the identity matrix. The middle plots display the residuals, which range between $-0.03$ and $0.03$, much smaller than the diagonal values. On the right, we show the distributions of the singular values, which are concentrated around 1.0, indicating that $W$ is close to being orthogonal and near the identity matrix.

The original CLIP model outputs normalized embeddings. Using an arbitrary linear projection can change the norm of the embeddings and hurt model performance and robustness. However, in our typical training scenario, this is not much of an issue since our transformation is close to an orthogonal transformation.

$$\|Wx\|_2^2 = x^\top W^\top W x \le \|x\|_2^2 \|W^\top W\| = \|W^\top W\| = \|W^\top W - I + I\|$$
$$\le \|W^\top W - I\| + \|I\| = 1 + \|W^\top W - I\| \approx 1 \Rightarrow \frac{Wx}{\|Wx\|} \approx Wx \quad (36)$$

However, when we amplify the singular values using matrix exponents, as discussed in Section 3.4, this difference grows exponentially; therefore, we should be careful and apply normalization.

## F    DATASETS & BASELINES

While human-curated preference sets are ideal, our methodology can still be effectively applied by designing the pretraining task in a semi-supervised manner. In both of our experiments, we did not rely on publicly available preference sets or human-curated preferences. Instead, we designed them ourselves, tailored to the specific task we wanted to fine-tune on. For instance, in the first experiment, we generated synthetic typographic attacks and used the original and targeted labels as preference labels. For debiasing the model, we used a dataset of images showing individuals of each gender performing various tasks and fine-tuned the model on an auxiliary task of reversing the model's gender understanding, again without curated preferences. This demonstrates that preference optimization can be effectively applied using task-specific, semi-supervised strategies.

### F.1    DATASETS

Table 3 presents the statistics of the datasets used in this paper. To evaluate the classification accuracy of our method on both original and typographic images (results in Table 1), we consider 9 datasets: ImageNet-100 (Deng et al., 2009), Caltech101 (Fei-Fei et al., 2004), OxfordPets (Parkhi et al., 2012), StanfordCars (Krause et al., 2013), Flowers102 (Nilsback & Zisserman, 2008), FGV-CAirCrafts (Maji et al., 2013), DTD (Cimpoi et al., 2014), SUN397 (Xiao et al., 2016) and EuroSAT (Helber et al., 2019). These datasets are not typographic attack datasets, so we follow the procedure in (Azuma & Matsui, 2023) to generate typographic attack images from these datasets. We also consider three real-world datasets designed for typographic attacks in Table 2: Materzynska (Materzynska et al., 2022), PAINT (Ilharco et al., 2022), and RTA-100 (Azuma & Matsui, 2023). Figure 10 shows some samples from the RTA-100 dataset. Additionally, for the Gender Bias results and analysis, we considered the VL-Bias (Zhang et al., 2022) dataset. Also, we consider Food101 (Bossard et al., 2014) for additional experiments, including saliency maps in Section C.

Table 2: Classification accuracy on real-world typographic datasets.

| Method | Materzynska | PAINT | RTA-100 | Avg. |
|---|---|---|---|---|
| CLIP | 43.27 | 50.00 | 47.20 | 46.82 |
| Materzynska+ | 77.78 | 55.45 | 57.60 | 63.61 |
| PAINT | 53.22 | 58.18 | 53.60 | 55.00 |
| Defense-Prefix | 71.93 | **63.64** | 58.00 | 64.52 |
| Ours (DPO) | 78.57 | 57.29 | 60.35 | 65.40 |
| Ours (IPO) | 77.98 | 59.38 | **65.27** | **67.54** |
| Ours (KTO) | **80.36** | 56.25 | 62.19 | 66.27 |

### F.2    BASELINES

In the typographic attack experiments, we used the following works as our baseline.

- (Materzynska et al., 2022). focus on disentangling the representation of written words and visual concepts in CLIP's image encoder. They achieve this through orthogonal projections, creating subspaces that isolate or eliminate the model's ability to process text. This

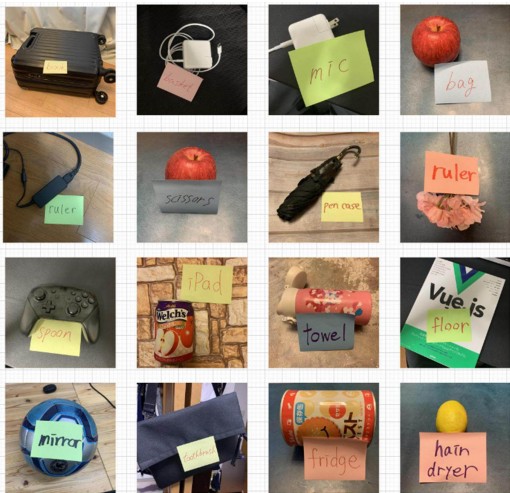

Figure 10: Samples of the RTA-100 dataset.

Table 3: Datasets statistics.

| Dataset | Number of Classes | Number of Images |
|---|---|---|
| ImageNet-100 | 100 | 130,000 |
| Caltech101 | 101 | 9,146 |
| OxfordPets | 37 | 7,393 |
| StanfordCars | 196 | 16,185 |
| Flowers102 | 102 | 8,189 |
| FGVCAircraft | 100 | 10,000 |
| DTD | 47 | 5,640 |
| SUN397 | 397 | 108,754 |
| EuroSAT | 10 | 27,000 |
| Food101 | 101 | 101,000 |
| Materzynska | 19 | 171 |
| PAINT | 89 | 110 |
| RTA-100 | 100 | 1,000 |
| VL-Bias | 65 | 24,000 |

disentanglement helps reduce text artifacts in image generation and offers defense against typographic attacks.

- (Ilharco et al., 2022) introduce PAINT (Patching with Interpolation), a method that linearly interpolates between the weights of a model before and after fine-tuning on a specific patching task ($\theta_{\text{patch}} = \alpha\theta_{\text{zs}} + (1-\alpha)\theta_{\text{ft}}$). This approach improves accuracy on tasks where the zero-shot CLIP model performs poorly, while preserving performance on tasks like ImageNet. PAINT is computationally efficient, as it doesn't require additional parameters or retraining from scratch. It also exhibits "broad transfer," where patching on one task can improve accuracy on related tasks, even with different classes.

- (Azuma & Matsui, 2023) present Defense-Prefix (DP), a technique that inserts a DP token before a class name in text prompts to make it resilient to typographic attacks. This method focuses on modifying the text input to CLIP rather than changing the model itself. DP significantly enhances accuracy on typographic attack datasets while maintaining the model's zero-shot capabilities. The authors also demonstrate its applicability to downstream tasks like object detection, where it effectively reduces the impact of typographic attacks without requiring additional training.

Table 1 does not include a comparison of our method with other baselines on the ImageNet dataset. This is because the experimental results of other methods were conducted on the ImageNetV2 validation subset, which was not publicly available at the time. Instead, we present results on a different subset of the ImageNetV2 dataset, referred to as the "*matched-frequency*" subset (Recht et al., 2019). Table 4 provides a comparison of our method with the pretrained CLIP model under the typographic attack scenario.

Table 4: Classification accuracy results on the ImageNetV2 dataset.

| Method | O | T |
| --- | --- | --- |
| CLIP | 54.21 | 33.23 |
| Ours (DPO) | **53.80** | 49.18 |
| Ours (IPO) | 47.61 | 42.75 |
| Ours (KTO) | 53.17 | **50.35** |

## G  HYPERPARAMETER SENSITIVITY ANALYSIS

In this section, we analyze the impact of several hyper-parameters such as $\beta$ and $\lambda$. We utilize the *FOOD101* dataset for training and *SUN* as the zero-shot dataset to assess model generalization. All experiments are conducted for 3 epochs, with a learning rate of $2 \times 10^{-5}$, and a coefficient $\gamma = 0.7$ in the Beta Moving Average.

### G.1  EFFECT OF $\beta$

The hyper-parameter $\beta$ controls the deviation from the policy in the loss function in Eq. (14). To assess its impact, we experiment with values between 0.01 and 1.5 on both the in-domain and zero-shot datasets.

As shown in Figure 11, our results indicate that IPO is highly sensitive to $\beta$. Additionally, the performance of both DPO and KTO decreases for excessively large $\beta$ values on the zero-shot dataset.

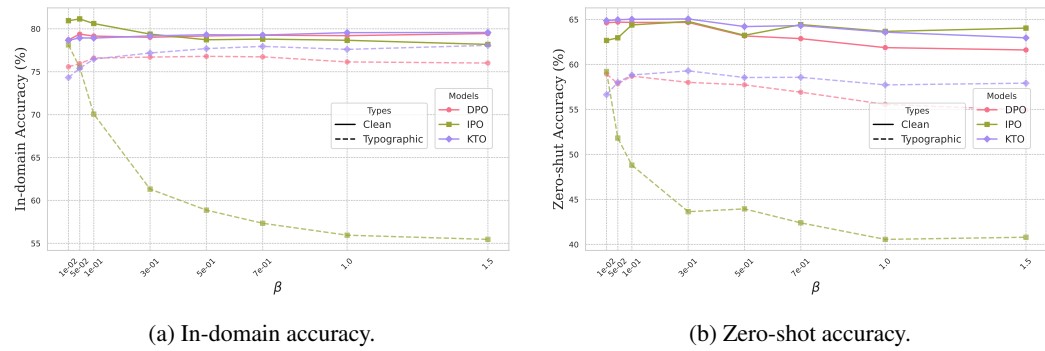

(a) In-domain accuracy.                    (b) Zero-shot accuracy.

Figure 11: Ablation study on the effect of the hyper-parameter $\beta$ on classification accuracies for clean and typographic datasets. The results highlight the performance of DPO, IPO, and KTO models and their impact on in-domain and zero-shot accuracy.

### G.2  ABLATION STUDY ON THE REGULARIZATION LOSS $\mathcal{L}_{\text{REG}}$ AND THE EFFECT OF $\lambda$

The hyperparameter $\lambda$ serves as the weight of the regularization term in our loss function. This regularizer is crucial for maintaining model performance on clean datasets during debiasing and adversarial training procedures. As shown in Figure 12, we vary $\lambda$ between 0.01 and 2.0 to analyze its impact on both in-domain and zero-shot datasets, highlighting a trade-off. In the zero-shot dataset, excessively large $\lambda$ values result in improved performance on the clean dataset but a decrease in performance on the typographic dataset. Conversely, very small values of $\lambda$ (e.g., 0.01) lead to lower performance on the clean dataset and higher performance on the typographic dataset.

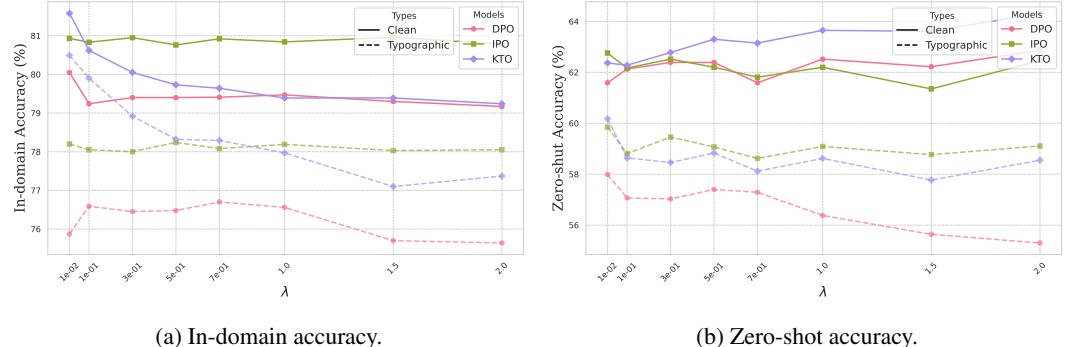

(a) In-domain accuracy.            (b) Zero-shot accuracy.

Figure 12: Ablation study on the effect of the hyper-parameter $\lambda$ on classification accuracies for clean and typographic datasets. The results highlight the performance of DPO, IPO, and KTO models and their impact in-domain and zero-shot accuracy.

## H   BETA MOVING AVERAGE VS. EXPONENTIAL MOVING AVERAGE

Fine-tuning enhances task-specific performance but can degrade a model's generalization, particularly in zero-shot tasks. Exponential Moving Average (EMA) tends to accelerate this degradation by diminishing the pre-trained model's influence over time, whereas Beta Moving Average (BMA) (Bayer et al., 2018) retains more knowledge from the pre-trained model, achieving a better balance between task-specific adaptation and generalization. This balance is essential for tasks like debiasing, adversarial robustness, and zero-shot classification, as evidenced in VLMs for out-of-distribution detection (Shu et al., 2023). BMA applies a temporal ensemble approach, weighting each training checkpoint along the fine-tuning trajectory by a Beta distribution to compute the final model state.

$$\theta^{\text{TE}} = \sum_{t=0}^{T} \frac{\alpha_t}{\sum_{k=0}^{T} \alpha_k} \cdot \theta_t, \tag{37}$$

where $\alpha_t$ is determined by a Beta distribution and controls how much influence each model has in the final ensemble. Unlike EMA, which rapidly diminishes the contribution of earlier models (including the pre-trained model), BMA ensures that $\theta_0$ remains a significant part of the model's knowledge, promoting better zero-shot generalization.

The weights $\alpha_t$ are drawn from the Beta distribution, normalized over the training steps:

$$\alpha_t = \text{Beta}(\gamma, \gamma) \left( \frac{t + 0.5}{T + 1} \right), \tag{38}$$

where $\gamma$ is a hyper-parameter that controls the balance between the pre-trained and fine-tuned models. By setting $\gamma < 1$, we ensure that both the pre-trained and fine-tuned models contribute more heavily to the final averaged model, thus reducing the forgetting problem of EMA.

To make BMA efficient in practice, it can be implemented as a moving average:

$$\theta_t^{\text{BMA}} = \frac{\sum_{k=0}^{t-1} \alpha_k \cdot \theta_k^{\text{BMA}} + \alpha_t \cdot \theta_t}{\sum_{k=0}^{t} \alpha_k}, \tag{39}$$

which allows us to compute the model's temporal ensemble without needing to store all the intermediate checkpoints.

In all of our reported results, we employed the BMA update strategy to enhance zero-shot performance. The specifics of our training procedure are detailed in Algorithm 2, where: $x$ denotes the original image, $\tilde{x}$ the perturbed image (e.g., typographic image), $y$ the label, and $\tilde{y}$ the adversarial label (e.g., typographic label). As demonstrated in Table 5, we compare the performance of BMA and EMA. The hyperparameters $\beta$ and $\lambda$ for each method are the same as those used during their training on ImageNet.

Table 5: Comparison of the effect of BMA and EMA optimization strategies on the in-domain dataset (FOOD101) and the zero-shot dataset (SUN).

(a) In-domain accuracy

| Method | Original | | Typographic | |
|---|---|---|---|---|
| | BMA | EMA | BMA | EMA |
| CLIP | 78.88 | | 51.43 | |
| DPO | 79.20 | 78.99 | 76.14 | 75.49 |
| IPO | 81.94 | 80.67 | 78.12 | 76.90 |
| KTO | 81.58 | 79.99 | 80.49 | 80.63 |

(b) Zero-shot accuracy

| Method | Original | | Typographic | |
|---|---|---|---|---|
| | BMA | EMA | BMA | EMA |
| CLIP | 61.00 | | 35.33 | |
| DPO | 61.87 | 62.26 | 55.56 | 54.08 |
| IPO | 62.67 | 61.85 | 59.22 | 58.22 |
| KTO | 62.37 | 61.76 | 60.18 | 59.20 |

---

**Algorithm 2** Preference optimization for contrastive learning with BMA details

---

**Require:** Pre-trained CLIP model $\pi_{\theta_0}$, Dataset $\mathcal{D} = (\mathcal{D}_{\text{pref}}, \mathcal{D}_{\text{reg}})$, Regularization coef. $\lambda_{\text{reg}}$, learning rate $\eta$
1: Initialize BMA model: $\theta_0^{\text{BMA}} \leftarrow \text{detach}(\theta_0)$
2: Initialize policy: $\theta_0^{\pi} \leftarrow \theta_0$
3: **for** $t = 1$ to $T$ **do**
4:      $b_{\text{pref}}, b_{\text{reg}} \leftarrow b = \{(x, y, \tilde{x}, \tilde{y})\}$          ▷ Get batch of preference / regularization data
5:      $l_{\text{pref}} \leftarrow \mathcal{L}_{\text{po}}(\pi_{\theta_{t-1}}, \pi_{\text{ref}}; b_{\text{pref}})$      ▷ Compute preference loss using $b_{\text{pref}} = \{(\tilde{x}, y, \tilde{y})\}$
6:      $l_{\text{reg}} \leftarrow \mathcal{L}_{\text{reg}}(\pi_{\theta_{t-1}}, \pi_{\text{ref}}; b_{\text{reg}})$      ▷ Compute regularization loss using $b_{\text{reg}} = \{(x, \tilde{x})\}$
7:      $l_{\text{tot}} \leftarrow l_{\text{reg}} + \lambda_{\text{reg}} \cdot l_{\text{reg}}$                             ▷ Total loss
8:      Update parameters of policy: $\theta_t^{\pi} \leftarrow \theta_{t-1}^{\pi} - \eta \nabla_{\theta_{t-1}^{\pi}} l_{\text{tot}}$
9:      Calculate $\alpha_t$ of the current model as in Eq. (38)
10:     Update the BMA model $\theta_t^{\text{BMA}}$ as in Eq. (39)
11: **end for**
**Ensure:** The final BMA model $\theta_T^{\text{BMA}}$
where: $x$ = Original image, $\tilde{x}$ = Typographic image, $y$ = Label, $\tilde{y}$ = Typographic label

---

# I SAMPLE EFFICIENCY ANALYSIS

In this section, we examine how efficiently our model learns with different dataset sizes. This analysis helps us understand how well the models perform with less data and how their accuracy changes as the dataset grows (see Figure 13). As expected, based on the foundation in Section 4.2 of (Gheshlaghi Azar et al., 2024), when there is insufficient training data, DPO tends to overfit, leading to poor performance on both *In-domain* and *Zero-shot* datasets, while KTO performs better in this situation.

Specifically, we focus on the impact of the number of typographic instances per image and how this affects both the in-domain and zero-shot generalization performance.

Table 6: Impact of number of typographic instances per image on in-domain (FOOD101) and zero-shot (SUN) performance. Results show accuracy (%) for different settings.

| Number of Instances | FOOD101 (In-domain) | | SUN (Zero-shot) | |
|---|---|---|---|---|
| | BMA | EMA | BMA | EMA |
| 1 | 74.85 | 75.24 | 58.27 | 57.47 |
| 3 | 75.22 | 75.25 | 58.83 | 58.46 |
| 5 | 75.59 | 75.49 | 58.18 | 58.07 |
| 10 | 76.25 | 75.78 | 57.88 | 57.29 |

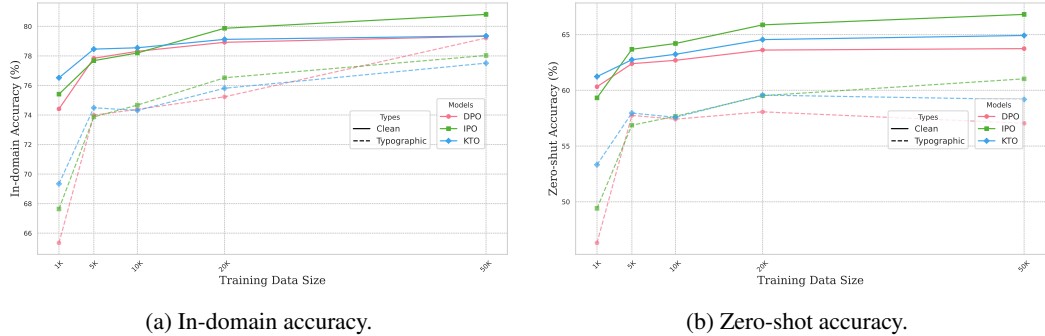

(a) In-domain accuracy.          (b) Zero-shot accuracy.

Figure 13: Ablation study on the effect of dataset size on classification accuracies for clean and typographic datasets. The results highlight the performance of DPO, IPO, and KTO models and their impact on in-domain and zero-shot accuracy.

## I.1 IMPACT OF NUMBER OF TYPOGRAPHIC INSTANCES PER IMAGE

To investigate the effect of varying the number of typographic instances per image, we conducted experiments using 1, 3, 5, and 10 typographic variations for each image on the *FOOD101* dataset (in-domain) and the *SUN* dataset (zero-shot). We evaluate the effect of varying the number of typographic instances per image on model performance. The results, summarized in Table 6, suggest that increasing the number of instances does not significantly improve accuracy, indicating that this augmentation technique may not be effective for enhancing the model's generalization further.

## J   PERFORMANCE COMPARISON OF DPO, IPO, AND KTO

In this section, we provide a detailed analysis of the performance of DPO, IPO, and KTO in our framework. The comparison focuses on key aspects such as sample efficiency, sensitivity to the $\beta$ hyperparameter, and performance across different datasets.

## J.1   SAMPLE EFFICIENCY

As discussed in Section I and illustrated in Figure 13, KTO demonstrates superior performance in scenarios where training data is scarce. This result aligns with our expectations, as KTO transforms each preference triplet $(x, y_w, y_l) \in \mathcal{D}_{\text{pref}}$ into two samples: $(x, y_{\text{desired}})$ and $(x, y_{\text{undesired}})$. By effectively doubling the number of training samples, KTO achieves greater sample efficiency, making it particularly beneficial in data-constrained settings.

In contrast, DPO performs the poorest among the three methods when training data is insufficient. This is consistent with the findings of (Gheshlaghi Azar et al., 2024), where DPO's tendency to overfit in low-data regimes is highlighted. Overfitting leads to poor generalization, resulting in sub-optimal performance on both *in-domain* and *zero-shot* datasets. IPO, while not as sample-efficient as KTO, outperforms DPO in such scenarios due to its relatively balanced approach to learning from preference data.

## J.2   SENSITIVITY TO $\beta$

Figure 11 highlights the sensitivity of each method to the $\beta$ hyperparameter, which governs the strength of the policy regularization. IPO is particularly sensitive to variations in $\beta$. Large $\beta$ values lead to a uniform policy distribution $\pi_\theta$, causing the model to assign significant probabilities to all actions. This behavior reduces specificity and harms performance, particularly on zero-shot datasets.

For both DPO and KTO, excessively high $\beta$ values similarly degrade performance. However, KTO exhibits slightly greater robustness compared to DPO in this regard, likely due to its ability to leverage additional training samples. To achieve reasonable performance with IPO, stronger constraints are required to retain the policy close to the reference model, as noted by (Gheshlaghi Azar et al., 2024). Fine-tuning $\beta$ is therefore crucial for optimizing IPO performance.

### J.3 Performance Across Different Datasets

Our ablation results in Figure 13 and the reported results in Table 1 indicate that with sufficient training data, all three methods—DPO, IPO, and KTO—exhibit comparable performance on both original and adversarial datasets. This convergence in performance is expected, as the final loss function in all three methods is derived from the same main objective (Eq. (3)), albeit with different design principles and optimization approaches. However, under specific conditions, their distinct characteristics lead to notable differences, highlighting the unique strengths and limitations of each method.

Nonetheless, specific strengths emerge in certain contexts:

- **Synthetic Datasets:** As shown in Table 1, KTO outperforms DPO and IPO on synthetic datasets, achieving better average performance on both the original and typographic variants.
- **Real-World Datasets:** As shown in Table 2, IPO achieves the best average performance among the three methods on real-world datasets.

### J.4 Weight Analysis of Different Variants

Regarding the specific analysis of the weight of different variants according to Eq. (10), we provide the following explanation:

- **DPO**: The gradient weight is given by:

$$w_{\text{pref}}(y_w, y_l; x) = \sigma(-\beta h_{\pi_\theta}(y_w, y_l, x)),$$

  where $\sigma(\cdot)$ is the sigmoid function. As $-\beta h_{\pi_\theta}(y_w, y_l, x)$ increases, the weight decreases, which results in lower emphasis on inputs with higher values of $h_{\pi_\theta}$, as they are already good enough. This deemphasizing prevents the model from straying too far from the base model.

- **IPO**: The gradient weight is defined as:

$$w_{\text{pref}}(y_w, y_l; x) = \left( \frac{1}{2\beta} - h_{\pi_\theta}(y_w, y_l, x) \right).$$

  Here, as $h_{\pi_\theta}(y_w, y_l, x)$ increases, the weight linearly decreases, becoming zero at $h_{\pi_\theta}(y_w, y_l, x) = \frac{1}{2\beta}$. In effect, this would stop the model from straying further than $\frac{1}{2\beta}$, ideally optimizing to around $h_{\pi_\theta}(y_w, y_l, x) \approx \frac{1}{2\beta}$.

- **Other Optimization Schemes**: Any other preference optimization scheme that uses the differential reward can be studied. One such case could be ROPO, as introduced by (Liang et al., 2024b).

We compare the different weighting schemes in Figure 14.

Figure 15 illustrates how the $h$ function correlates with dataset difficulty, highlighting the relationship between model performance and the inherent challenges posed by different datasets.

## K Experiments on Gradient-based Attacks

In this experiment, we evaluate the effectiveness of our proposed method in enhancing model robustness against gradient-based attacks, specifically the Projected Gradient Descent (PGD) attack. PGD is a widely used method for generating adversarial examples by iteratively perturbing the input to maximize the model's loss.

The setup is as follows: we use targeted PGD attacks to create an image $x'$, which the model predicts the caption $y_l$ for instead of $y_w$, where $y_l$ is the target caption of the adversary. Our methodology from the typographic attack section remains largely the same, with a few adjustments. Specifically, we attack the model in an online fashion during training. The training accuracies of the model

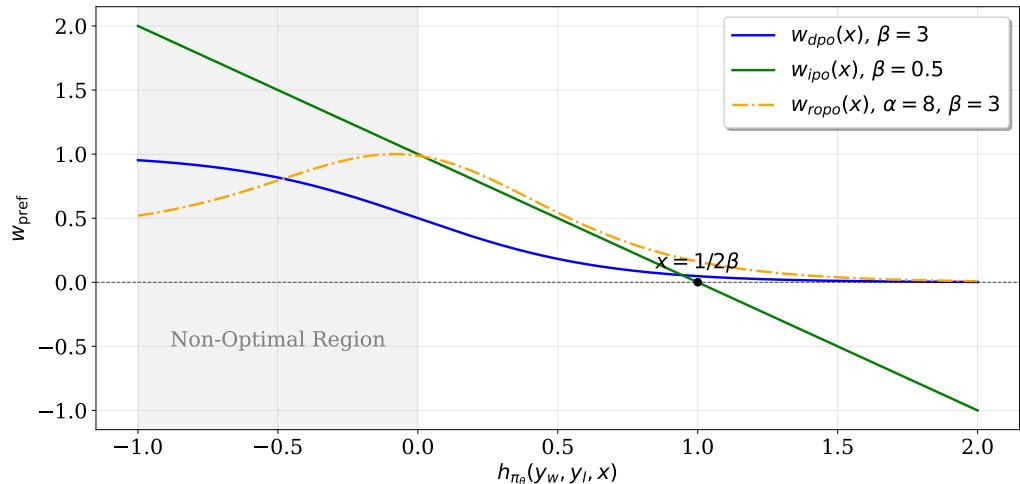

Figure 14: Visualization of different weighting schemes.

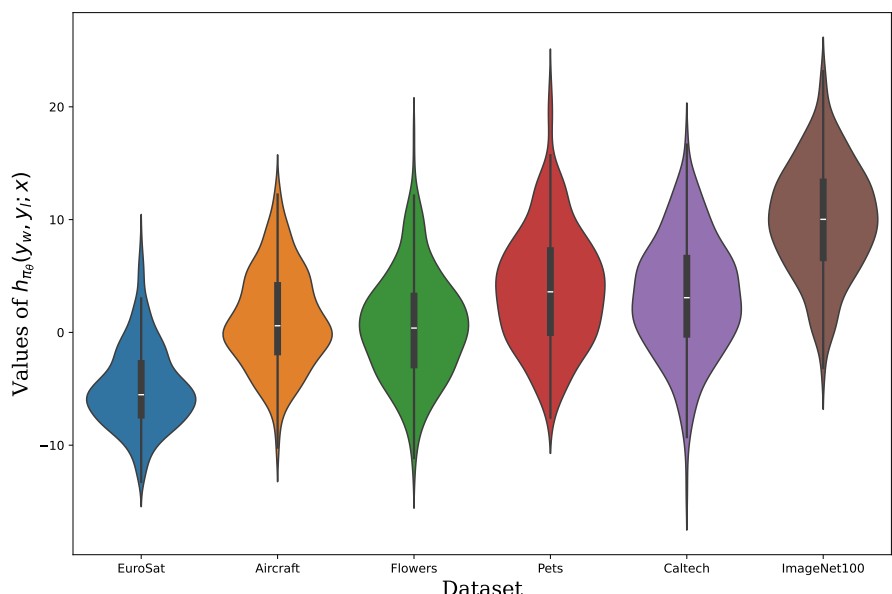

Figure 15: The violin plot represents the distribution of the $h$ function in DPO method across different datasets. From left to right, the datasets progressively become less challenging. By 'challenging,' we refer to the performance of the pretrained model on the typographic variation of each dataset. As seen, the dataset on the far right corresponds to our training set, which shows higher and more accurate $h$ values.

on clean and perturbed data are illustrated in Figures 16 and 17. Additionally, our results on the validation set of the dataset are provided in Table 7.

In these experiments, for $\epsilon = 4/255$, we used 700 iterations with a batch size of 256, and for $\epsilon = 2/255$, we used 1000 iterations with a batch size of 256.

Overall, it is evident that the clean data accuracy remains the same while adversarial robustness is increasing.

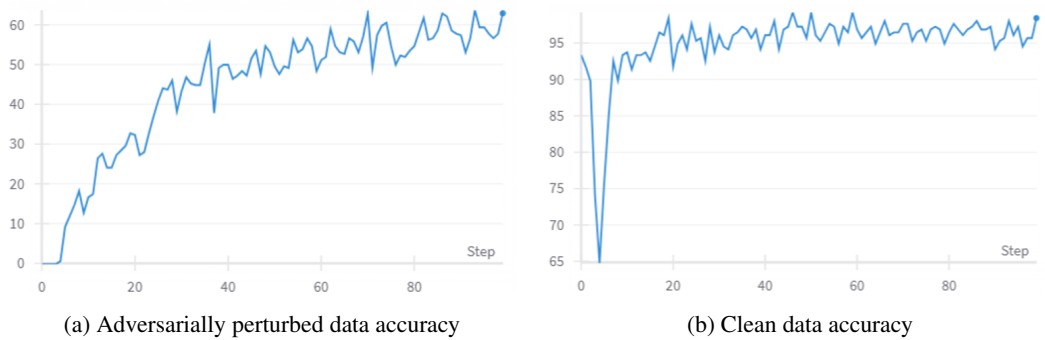

(a) Adversarially perturbed data accuracy                    (b) Clean data accuracy

Figure 16: Results of training on adversarial robustness, CIFAR10 dataset, $\epsilon = 2/255$.

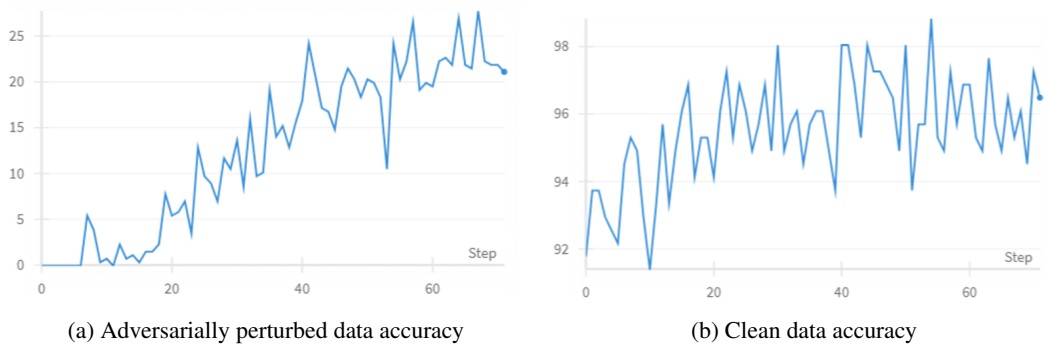

(a) Adversarially perturbed data accuracy                    (b) Clean data accuracy

Figure 17: Results of training on adversarial robustness, CIFAR10 dataset, $\epsilon = 4/255$.

| Dataset | Epsilon ($\epsilon$) | Accuracy on Adversarial Images | Accuracy on Clean Images |
|---------|---------|---------|---------|
| CIFAR-10 | $2/255$ | 62.89% | 98.44% |
| CIFAR-10 | $4/255$ | 21.88% | 97.27% |

Table 7: Evaluation of Model Accuracy on CIFAR-10 Dataset for Clean and Adversarially Perturbed Images.

