# OpenReview forum: "Aligning Visual Contrastive learning models via Preference Optimization"
_ICLR.cc/2025/Conference — ICLR 2025 Poster_

### Official Review · Reviewer_y7hh · 2024-10-25

**Soundness:** 2
**Presentation:** 1
**Contribution:** 2
**Rating:** 3
**Confidence:** 4

**Summary:**

This paper introduces Preference Optimization for training the contrastive learning model CLIP, aiming to enhance the model's robustness against typographic attacks and mitigate gender biases. This approach aligns the model with human preferences. Experimental results on datasets such as ImageNet, Caltech101, and OxfordPets demonstrate the effectiveness of this method.

**Strengths:**

It is significant to explore aligning non-generative model with human preferences using Preference Optimization.

This paper is well-motivated.

**Weaknesses:**

See questions.

**Questions:**

1. Could this method be applied to other tasks apart from enhancing robustness against typographic attacks and mitigating gender biases?
2. Could you provide the ablation study results for components $\mathcal{L}_{pref}$ and $\mathcal{L}_{reg}$ in the loss function?
3. I do not understand the image in the left part of Figure 1. What does the obscured dog in the left part of Figure 1 signify?
4. What is the role of section 3.4 in your method? Why is fine-tuning the model mentioned in Section 3.4?
5. Could you provide evidence for your claim “the overall matrix $W^TW$ remains close to the identity matrix” in Section3.4？
6. Section 3 does not clearly introduce the method. In the last part of Section 3, you should package and summarize your method to give readers an overall understanding.
7. In Table 1, although the accuracy on the typographic dataset has increased compared to other methods, the accuracy on the original dataset has generally decreased. Therefore, the method has harmed the pretrained knowledge.
8. Could you provide an analysis of the different performances of DPO, IPO, and KTO in your method in Section 4?
9. What is the relationship between Section 4.4.1 and Section 4.1? I am not sure about the role of Section 4.4.1.
10.  What is the meaning of transformation scaling t in Section 4?

---

> ### Author Response · Authors · 2024-11-24
> **Thanks for your careful review (Part 1/3)**
>
> # Response to Reviewer Comments
>
> ## Q1: Generalizability of the Proposed Method
> > Could this method be applied to other tasks apart from enhancing robustness against typographic attacks and mitigating gender biases?
>
> Thank you for your thoughtful question. As we mentioned in the paper, the proposed method is not limited to enhancing robustness against typographic attacks or mitigating gender biases. It is a general framework that can be applied in any situation where we aim to optimize a model's behavior toward a preferred target while distancing it from a dispreferred target.
>
> For instance, this approach can be effective in addressing general targeted adversarial attacks. In this scenario, the attacker seeks to trick the model into choosing an adversarial class $y_l$ instead of the true class $y_w$. Our method allows explicit control over the model's behavior in such cases by leveraging preference-based optimization to reduce the probability of the adversarial class while increasing the likelihood of the true class.
>
> More broadly, this framework can be extended to tasks involving $\textit{preference-based comparisons}$ between two behaviors or distributions. In practice, such comparisons rely on samples from the distributions, allowing us to explicitly align the model with desired behaviors. This makes the method highly versatile for applications where fairness, robustness, or alignment with specific human values is required.
>
> ---
>
> ## Q2: Ablation Study Results for Loss Function Components
> > Could you provide the ablation study results for components $\mathcal{L}_\text{pre}$ and $\mathcal{L}_\text{reg}$ in the loss function?
>
> As detailed in Appendix F.2 (Appendix G.2 of the revised version), we conducted an ablation study to analyze the contribution of each component in the loss function, specifically, preference loss and regularization loss, on both in-domain and zero-shot datasets. By varying the $\lambda_\text{reg}$ coefficient from 0.01 to 2, we demonstrated the trade-off between retaining pretrained knowledge and achieving robustness against typographic attacks.
>
> As illustrated in Figure 10, excessively large $\lambda_\text{reg}$ values lead to improved performance on the clean dataset but cause a decrease in performance on the typographic dataset. Conversely, very small $\lambda_\text{reg}$ values (e.g., 0.01) result in reduced performance on the clean dataset while improving robustness against typographic attacks. This highlights the expected trade-off, making $\lambda_\text{reg}$ a critical hyperparameter for balancing clean data accuracy and adversarial robustness in our framework.
>
> ---
>
> ## Q3: Explanation of the Left part of Figure 1
> > I do not understand the image in the left part of Figure 1. What does the obscured dog in the left part of Figure 1 signify?
>
> The left dog in Figure 1 signifies a data sample belonging to $\mathcal{D}_\text{pref}$. As we mentioned in Section 3.2, each sample in the preference dataset consists of a triplet $(x', y_w, y_l)$, where $x'$ is the image, and $y_w$ and $y_l$ are the preferred and dispreferred labels, respectively.
>
> Regarding the image $x'$, the goal is to control the model's behavior for this specific input. For instance, in the task of enhancing robustness against typographic attacks, this input image $(x')$ represents a typographic image—e.g., an image of a dog with the word $\textit{"cat"}$ written on it. More generally, $x'$ can represent a perturbed (adversarial) image generated under any adversarial attack. The preference dataset $\mathcal{D}_\text{pref}$ plays a critical role in teaching the model to distinguish between desired and undesired outputs for such challenging inputs.

---

> ### Author Response · Authors · 2024-11-24
> **Thanks for your careful review (Part 2/3)**
>
> ## Q4: The role of Section 3.4
> > What is the role of Section 3.4 in your method? Why is fine-tuning the model mentioned in Section 3.4?
>
> In Section 3.3, we detailed our framework. This framework can be applied in two ways: by fine-tuning the entire set of model parameters or by fine-tuning a linear projection layer applied to both the text and image encoders. Section 3.4 explains the concept of learning a shared linear layer on top of both encoders. Specifically, we introduce a learnable matrix as a shared projection head for both the text and image encoders and this matrix is optimized using the same method described in Section 3.3.
>
> One of the key contributions of Section 3.4 is demonstrating how the Singular Value Decomposition (SVD) of this learnable matrix is interpretable and enables fine-grained control over the model’s behavior. By decomposing the matrix into its singular values and singular vectors, we show that adjusting the singular values allows us to control (increase or decrease) specific aspects of the model’s behavior. For example, this approach can even reverse a learned aspect of the model, e.g. the trade-off between OCR and object detection as provided in Figure 2.
>
> This method adds flexibility by enabling post-training adjustments to the model without requiring retraining. It complements the broader framework described in Section 3.3 by offering an interpretable and computationally efficient way to adapt the model's behavior in alignment with specific preferences.
>
>
> ---
>
> ## Q5: Evidence that the $W^{T}W$ Matrix is Close to Identity
> > Could you provide evidence for your claim “the overall matrix $W^{T}W$ remains close to the identity matrix” in Section 3.4?
>
> We appreciate your attention to this detail. We have addressed this point by providing additional analysis and experimental evidence to support the claim that “the overall matrix $W^{T}W$ remains close to the identity matrix”. The detailed results, including quantitative evidence and visualizations, are now included in Appendix E. We hope this additional analysis clarifies and substantiates our claim.
>
> ---
>
> ## Q6: Clarifying Section 3 Summary
> > Section 3 does not clearly introduce the method. In the last part of Section 3, you should package and summarize your method to give readers an overall understanding.
>
> Thank you for your valuable feedback. We have addressed your comment by adding a concise summary and a short conclusion at the end of Section 3.3 to provide readers with a clear overview of our proposed method. Additionally, we have included Algorithm 1 to further clarify and illustrate the method.
>
> ---
>
> ## Q7: Typographic Datasets Accuracy
> > In Table 1, although the accuracy on the typographic dataset has increased compared to other methods, the accuracy on the original dataset has generally decreased. Therefore, the method has harmed the pretrained knowledge.
>
> Thank you for pointing this out. In the revised version of the paper, we have clarified this point in the last part of Appendix F. Specifically, the experimental results of other methods were conducted on the ImageNetV2 validation subset, which was not publicly available at the time of our experiments. Instead, we evaluated our method on a different subset of the ImageNetV2 dataset, referred to as the "matched-frequency" subset. Table 4 provides a detailed comparison of our method with the pretrained CLIP model under the typographic attack on the ImageNet dataset.
>
> Additionally, we have identified and addressed an issue in our presentation of Table 1. The KTO method significantly outperforms all baselines on the typographic dataset while performing nearly identically to the state-of-the-art (SOTA) method on the original dataset. We have revised Table 1 to make these distinctions clearer and provide a more accurate portrayal of the performance of our proposed methods across different datasets.
>
> ---
>
> ## Q8: Analysis of DPO, IPO, and KTO
>
> > Could you provide an analysis of the different performances of DPO, IPO, and KTO in your method in Section 4?
>
> Thank you for the valuable suggestion. In the earlier version of our paper, we conducted ablation studies in Appendices F and H (Appendices G and I in the revised version), where we compared and analyzed the performance of our proposed method under different scenarios. In the revised version, we have expanded this analysis and presented a more detailed and organized comparison of DPO, IPO, and KTO in Appendix J.

---

> ### Author Response · Authors · 2024-11-24
> **Thanks for your careful review (Part 3/3)**
>
> ## Q9: Relationship Between Sections 4.4.1 and 4.1
>
> > What is the relationship between Section 4.4.1 and Section 4.1? I am not sure about the role of Section 4.4.1.
>
> If you are referring to Section 4.1.1, we explored the effect of varying the transformation scaling $t$ to modify the model’s performance without retraining, solely by manipulating the spectrum of the learned matrix $W$. In contrast, Section 4.1 focuses on showcasing the performance of the final trained model across various datasets, without any manipulation of $t$.
>
> ---
>
> ## Q10: Meaning of Transformation Scaling $t$
> > What is the meaning of transformation scaling $t$ in Section 4?
>
> The transformation scaling $t$ in Section 4 refers to the exponent applied to the singular values as a post-processing method, allowing us to strengthen or weaken certain directions in the embedding space. We have added this clarification in the paper, in Section 3.4. Thank you for pointing this out.

---

> > ### Comment · Reviewer_y7hh · 2024-11-25
> >
> > Thank you for your detailed response. Most of my concerns have been addressed. Could you provide the experimental results of this method on other tasks for my question1? If the experimental results confirm its effectiveness on other tasks, I will increase score.

---

> > > ### Author Response · Authors · 2024-12-01
> > > **effectiveness on other tasks**
> > >
> > > Thank you for your valuable suggestion,
> > > Analyzing the model's effectiveness in more diverse tasks is crucial. To prove we have not cherry-picked these specific tasks, we have provided more analysis in our comments to all reviewers.

---

### Official Review · Reviewer_ck2S · 2024-10-27

**Soundness:** 3
**Presentation:** 2
**Contribution:** 3
**Rating:** 6
**Confidence:** 4

**Summary:**

The paper revisits well-known alignment techniques, such as Direct Preference Optimization (DPO) and Identity Preference Optimization (IPO), in the representation space learned by CLIP. The idea is simple yet effective: reformulate the policy $\pi$ in DPO and IPO by using the similarity scores between preference texts, $y_w$ (preferred) and $y_l$ (unpreferred), and the given adversarial image $x'$. The authors evaluate the proposed method on typographic attacks and show that it improves the CLIP model’s robustness to these attacks while preserving performance on the original datasets (without typographic attacks). To mitigate the overfitting issue of training large models on small datasets, the authors propose training a linear layer (parameterized by $W$) appended to the visual encoder, with both the pre-trained visual and text encoders frozen. Additionally, the authors propose applying SVD decomposition over $W$ as $W=U\Sigma^tV$, allowing the alignment magnitude to be controlled by $t\in\mathcal{R}$. The authors demonstrate that a learned alignment for gender bias can be effectively controlled by adjusting $t$.

**Strengths:**

1. The proposed method is simple yet effective.
2. The authors provide a new perspective on IPO and DPO concerning the representation space learned by CLIP.
3. The alignment controllability through $t$ is effective.
4. The background and motivation are well-organized.

**Weaknesses:**

1. Clarity needs improvement.
    * $\mathcal{L}_{pref}$ in (10) appears without a definition. In Corollary 3.2, it is assumed to be either the DPO loss or IPO loss, while the experiments further include the case of KTO loss.
    * In (9), $\mathcal{I}_{ref}$ is frozen and has no trainable parameters, contributing solely to per-example weighting when substituted in (5), (6), and (7). It is recommended to clarify this in advance.
    * In Fig.1, $\mathcal{L}_{pref}$ is computed with the given triplet $(y_w, y_l, x’)$, where $x’$ is an adversarial image. The presence of multiple negative text representations, such as $\tau_1$, $\tau_2$, and $\tau_3$, is confusing without specifying either $y_w$ or $y_l$ as text inputs.
    * The overall loss in (13) is iterated over two different datasets, $D_{pref}$ and $D_{reg}$, simultaneously. Further explanation is needed on how the inputs $(y_w, y_l, x’)\in D_{pref}$ and $x\in D_{reg}$ are paired or sampled.
    * The bottom row (differences) in Table 1 is confusing, and it cannot correctly demonstrate the trade-off between O (Original dataset) and T (Target dataset) for each variant of the proposed method. The reviewer recommends indicating the improvement or degradation alongside each accuracy as $\color{green}{(+1.0)}$ or $\color{red}{(-1.0)}$ relative to the base model, i.e., CLIP, for clarity.
2. Lack of a concrete conclusion over comparisons with baselines. The results in Table 1 deserve more discussion. Examples are listed below.
    * No method in Table 1 consistently outperforms the others. Is there a large domain gap between different datasets that prevents any method from generalizing well across all of them?
    * PAINT significantly outperforms the proposed method (including all variants: DPO, IPO, and KTO) on both O and T in the ImageNet* column. Is the constraint of a single trainable linear layer in the proposed method too restrictive?
    * A comparison between different variants of the proposed method would be valuable. For example, what types of inputs are weighted more in different variants according to (10)?

**Questions:**

The reviewer appreciates the well-organized background of DPO, IPO, and KTO, as well as the new perspective on these methods with CLIP. The paper focuses on providing new insights into preference optimization with CLIP. However, the most significant difference between DPO, IPO, and KTO—i.e., per-example weighting—is not discussed sufficiently in the paper (KTO might require further discussion). Additionally, the lack of a thorough comparison and the absence of clear distinctions in Table 1 together weaken the contribution. Therefore, the reviewer’s main questions are as follows:

1. What are the differences among the proposed method variants? Which types of inputs or datasets are weighted more or preferred for each variant?
2. Could these variants be unified into a single method that outperforms the other baselines in Table 1?

The reviewer is open to reconsidering the rating if the authors could address these questions (including those in Weakness section).

Some typos:
* In Ln. 023, “Our experiments” appears incorrectly inserted.
* Table 2 lacks a reference. The possible related reference is in Ln. 1073 in the appendix.
* Several papers in the references are duplicated: Ln. 551 and Ln. 554; Ln. 559 and Ln. 562; Ln. 681 and Ln. 687; Ln. 750 and Ln. 754.

---

> ### Author Response · Authors · 2024-11-25
> **Thanks for your careful review (Part 1/3)**
>
> # About the weaknesses
>
> >  $\mathcal{L}_{\text{pref}}$ in (10) appears without a definition. In Corollary 3.2, it is assumed to be either the DPO loss or IPO loss, while the experiments further include the case of KTO loss.
>
> Thank you for pointing this out. We have clarified in the updated text that $\mathcal{L}_{\text{pref}}$ refers to the DPO, IPO, or KTO loss, depending on the context, and ensured consistency between Corollary 3.2 and the experiments.
>
> > In (9), $\mathcal{I}_{\text{ref}}$ is frozen and has no trainable parameters, contributing solely to per-example weighting when substituted in (5), (6), and (7). It is recommended to clarify this in advance.
>
> Thank you for the suggestion. We have clarified in the updated text that $\mathcal{I}_{\text{ref}}$ is frozen and has no trainable parameters, contributing solely to per-example weighting when used in equations (5), (6), and (7).
>
> > In Fig. 1, $\mathcal{L}_\text{pref}$ is computed with the given triplet $(y_w, y_l, x')$, where $x'$ is an adversarial image.
> The presence of multiple negative text representations, such as $\tau_1$, $\tau_2$, and $\tau_3$, is confusing without specifying either $y_w$ or $y_l$ as text inputs.
>
> While the loss depends on $(y_w, y_l, x')$, the overall action set (the set of possible captions for that image) is still important. In the typical MDP scenario, the action set is constant and does not depend on the state (the input image). Therefore, in $\mathcal{L}_\text{pref}$, we use a constant action set like `an image of a [class]` using all available classes in the dataset, and then for each sample there are preferred and dispreferred labels $(y_w, y_l)$, corresponding to different actions.
>
> >  The overall loss in (13) is iterated over two different datasets, $D_\text{pref}$ and $D_\text{reg}$, simultaneously.
>     Further explanation is needed on how the inputs $(y_w, y_l, x') \in D_\text{pref}$ and $x \in D_\text{reg}$ are paired or sampled.
>
> Generally speaking, $D_\text{pref}$ and $D_\text{reg}$ can be very different datasets, so we would sample from each dataset in every iteration. However, in many cases, we can use similar datasets for the overall objective.
>
> For example, in our experiments regarding typographic attacks, $D_\text{reg}$ contained the clean images, and $D_\text{pref}$ contained the typographic images with the preferred and dispreferred labels. For this task, we used a composite dataset $D=(x, x', y_w, y_l) $, which contained all the required inputs, with $x'$ being a typographic sample generated from the clean image $x$.
>
> Regarding the gender debiasing task, both $D_\text{reg}$ and $D_\text{pref}$ utilized the same image dataset but with different action sets (the set of possible labels), where we used templates like "this person is doing [activity]" for $D_\text{reg}$ and "this [gender] is doing [activity]" for $D_\text{pref}$.
>
> Overall, one can combine the two datasets into a single overall dataset and iterate over that, or use completely different datasets.
>
> > The bottom row (differences) in Table 1 is confusing, and it cannot correctly demonstrate the trade-off between O (Original dataset) and T (Target dataset) for each variant of the proposed method. The reviewer recommends indicating the improvement or degradation alongside each accuracy as (+1.0) or (-1.0) relative to the base model, i.e., CLIP, for clarity.
>
> Thank you for highlighting this point. In the revised version of the paper, we clarified this issue in the final part of Appendix F. Notably, the experimental results of other methods were conducted on the ImageNetV2 validation subset, which was not publicly available during our experiments. Instead, we evaluated our method on the "matched-frequency" subset of the ImageNetV2 dataset. Table 4 now provides a detailed comparison between our method and the pretrained CLIP model under the typographic attack scenario on the ImageNet dataset.
>
> Furthermore, we identified and addressed an issue in the presentation of Table 1. The KTO method significantly outperforms all baselines on the typographic dataset while achieving performance nearly identical to the state-of-the-art (SOTA) method on the original dataset. Table 1 has been revised to clarify these distinctions and offer a more accurate representation of our proposed methods' performance across various datasets.

---

> ### Author Response · Authors · 2024-11-25
> **Thanks for your careful review (Part 2/3)**
>
> > No method in Table 1 consistently outperforms the others. Is there a large domain gap between different datasets that prevents any method from generalizing well across all of them?
>
> Thank you for your observation. After revising Table 1 to enhance clarity and removing the ImageNetV2 dataset (we will provide a detailed explanation for this in a separate comment to all authors), it becomes evident that our proposed methods (DPO, IPO, and KTO) consistently outperform all baselines on the typographic dataset. Furthermore, the KTO variant performs nearly identically to the state-of-the-art (SOTA) method on the original dataset. The revised table now highlights these distinctions more effectively, making the performance differences across datasets clearer.
>
> > PAINT significantly outperforms the proposed method (including all variants: DPO, IPO, and KTO) on both O and T in the ImageNet* column. Is the constraint of a single trainable linear layer in the proposed method too restrictive?
>
> We will explain our reasoning in a comment for all authors.
>
> > A comparison between different variants of the proposed method would be valuable. For example, what types of inputs are weighted more in different variants according to (10)?
>
> Thank you for your insightful suggestion. In the earlier version of our paper, we conducted ablation studies in Appendices F and H (now Appendices G and I in the revised version), where we compared and analyzed the performance of our proposed method under different scenarios. In the revised version, we have expanded this analysis and provided a more detailed and systematic comparison of DPO, IPO, and KTO in Appendix J.
>
> Regarding your specific point about analyzing the weight of different variants according to Equation (10), we provide the following explanation:
> - In the case of **DPO**, the gradient weight is given by:
>   $$w_{\text{pref}}(y_w, y_l; x) = \sigma(-\beta h_{\pi_\theta}(y_w, y_l, x)),$$
>
>   where $\sigma(\cdot)$ is the sigmoid function. As $-\beta h_{\pi_\theta}(y_w, y_l, x)$ increases, the weight decreases, which results in lower emphasis on inputs with higher values of $h_{\pi_\theta}$ , as they are already good enough. This deemphasizing prevents the model from straying too far from the base model.
>
> - For **IPO**, the gradient weight is defined as:
>   $$w_{\text{pref}}(y_w, y_l; x) = \left(\frac{1}{2\beta} - h_{\pi_\theta}(y_w, y_l, x)\right).$$
>
>   Here, as $h_{\pi_\theta}(y_w, y_l, x)$ increases, the weight linearly decreases, becoming zero at $h_{\pi_\theta}(y_w, y_l, x)=\frac{1}{2\beta}$. In effect, this would stop the model from straying further than $\frac{1}{2\beta}$, ideally optimizing to around $h_{\pi_\theta}(y_w, y_l, x)\approx\frac{1}{2\beta}$.
>
> Any other preference optimization scheme that uses the differential reward can be studied. One case could be ROPO, as introduced by [1]. We have added some plots for visualizing the different weighting schemes in Appendix J.4.
>
> [1] ize Liang, Chao Chen, Shuang Qiu, Jie Wang, Yue Wu, Zhihang Fu, Zhihao Shi, Feng Wu, and Jieping Ye. 2024. Ropo: Robust preference optimization for large language models. Preprint, arXiv:2404.04102.

---

> ### Author Response · Authors · 2024-11-25
> **Thanks for your careful review (Part 3/3)**
>
> # Answer to the questions
>
> > What are the differences among the proposed method variants? Which types of inputs or datasets are weighted more or preferred for each variant?
>
> In the previous version of our paper, we conducted ablation studies in Appendices F and H (Appendices G and I in the revised version), where we analyzed and compared the performance of our proposed method across various scenarios. In the updated version, we have expanded this analysis and provided a more detailed and organized comparison of DPO, IPO, and KTO in Appendix J. This revised section offers a thorough evaluation to better highlight the differences among the methods.
>
> To address the second part of your question, as discussed in Appendix J.3, our ablation results (Figure 12 and Table 1) indicate that with sufficient training data, all three methods—DPO, IPO, and KTO—perform comparably on both original and adversarial datasets. This is because the final loss function for all three methods is derived from the same main objective (Eq. 3), despite differences in their design principles and optimization approaches. However, under specific conditions, their distinct characteristics yield meaningful differences, which highlight their unique strengths and limitations.
>
> In summary:
>
>  - **Synthetic Datasets:** KTO has better average performance on synthetic datasets, including original and typographic variants.
>  - **Real-World Datasets:** IPO achieves the highest average performance among the three methods on real-world datasets, as shown in Table 2.
>
> > Could these variants be unified into a single method that outperforms the other baselines in Table 1?
>
> Thank you for this suggestion. Unifying these variants into a single method could potentially outperform the baselines in Table 1. This could involve combining their strengths through multi-objective optimization, incorporating both preference alignment and robustness terms, or using adaptive weighting and hierarchical training strategies. However, careful balancing and further experimentation and ablation studies would be necessary to validate whether such a unified approach consistently outperforms the existing baselines across diverse settings.
>
> Additionally, we would like to state that after adding some clarity improvements to Table 1 and removing the ImageNetV2 dataset from the table (we will explain our reasoning in a comment for all authors), it is more clear that all variants of our method (DPO, IPO, KTO) significantly outperform all baselines on the typographic dataset and performs nearly identically to the state-of-the-art (SOTA) method on the original dataset. We have revised the table to make this distinction clearer.

---

> ### Comment · Reviewer_ck2S · 2024-11-26
>
> The Authors' responses to the concerns raised in the initial review are appreciated. While some issues have been addressed, there are still areas that require improvement. Based on the efforts to address the feedback and the potential of the work, the rating is raised to a weak accept as encouragement. The following points should be addressed in the revision:
>
> **Ambiguous Use of Mathematical Symbols**
>
> While the overall structure of the paper is understandable, the definitions of many symbols vary depending on the context, making it difficult to follow the details. This issue needs to be resolved to ensure clarity.
> * For instance, symbols like $L_{pref}$​ and $y_w$​ (in Fig. 1) are not well-defined. Simply notifying the ambiguity in Ln. 306 is not the correct way. The reviewer suggests a rearrangement, moving (14) to earlier section as a brief overview, and then discussing different cases such as DPO, ITO, and KTO.
> * Additionally, it is unclear whether the constant action set is also iterated through by $y_w$​.
>     The constant action set, which contributes to $L_{pref}$​, should be clearly defined in the main text, and this relationship should be explicitly noted in Fig. 1
>
> **Focus: Analysis or SOTA?**
>
> The Authors attempt to claim state-of-the-art (SOTA) performance, but this aspect is primarily attributed to KTO, which is not the main focus of the paper. Shifting the emphasis towards analysis would result in more meaningful discussions. For example, Fig. 13 in the Appendix brings meaningful discussions. Suggestions include:
>
> * Further develop Appendix J, adding experimental support for each subsection.
> * Consider a deeper exploration, such as linking $h_\pi$​ to dataset difficulty. The Focal Loss paper provides relevant examples for such demonstrations.
>
> Additionally, the claim about the sample efficiency of KTO (e.g., Ln. 1415) should be supported by references or additional experiments.
>
> **Improving Readability**
>
> Beyond the ambiguity of mathematical symbols, the paper's readability is hindered by frequent references to the Appendix, requiring readers to frequently jump between sections.
>
> For instance, three appendix references appear in a single line (e.g., Ln. 413). Critical content should be moved into the main paper, while less essential sections, such as Sec. 4.2 and Sec. 4.3, could be relocated to the Appendix to streamline the main text.

---

> > ### Author Response · Authors · 2024-12-01
> >
> > We agree that our paper provides a comparison of different preference optimization techniques, with KTO performing the best according to our evaluation. Therefore, we have slightly updated the abstract, contributions, and conclusion to emphasize this comparative study design. Additionally, we have moved some references to the appendix as footnotes.
> >
> > In the revised version of the paper, we clearly define $\mathcal{L}_\text{pref}$ in Line 313. Additionally, we use the notations $y_w$ and $y_l$ for preferred and dispreferred actions, respectively, as they are common in the literature of RLHF and preference optimization. We also define the notations $y_w$ and $y_l$ in a general sense in Line 188, and their specific usage in our setting is clarified in Line 306.
> >
> >
> > Regarding the further development of Appendix J, we provided experimental results to support the claims of each subsection (e.g., Figures 10 and 12). In Section J.3, we analyze the main results of the paper (Tables 1 and 2) and compare our different methods. Additionally, with respect to the relationship between $h_\pi$ and dataset difficulty, your suggestion was very useful and encouraged us to perform further analysis on this component of the loss function, as illustrated in Figure 14 and Appendix J.4.
> >
> > Furthermore, to demonstrate the versatility of our proposed framework and its effectiveness in diverse tasks, we perform new experiments, which are placed in Appendix K. Please refer to our official comment to all reviewers for the numerical values and more details on the experiments.

---

> ### Comment · Reviewer_ck2S · 2024-12-02
>
> The authors' responses are appreciated, and most concerns have been effectively addressed. Considering the authors' responses and other reviews, the positive score is maintained due to the demonstrated potential of this work.

---

### Official Review · Reviewer_PYYa · 2024-11-04

**Soundness:** 3
**Presentation:** 4
**Contribution:** 4
**Rating:** 8
**Confidence:** 2

**Summary:**

This paper introduces a method for aligning contrastive learning models (CLIP), with human preferences using preference optimization techniques such as DPO and IPO. By formulating contrastive learning as a one-step MDP and fine-tuning CLIP with these techniques, the authors enhance the model robustness of CLIP against typographic attacks and mitigate biases, particularly around gender. Experimental results show improvement on multiple datasets.

**Strengths:**

Originality: This is the first work to improve contrastive learning models through Preference Optimization. The idea of leveraging true labels and typographic labels for preferences, instead of curating a separate preference set from human annotation, is novel and interesting.

Clarity: This paper is well-written and has very clear motivations, backgrounds, methods, and experiments.

Significance: The topic of aligning human preferences in contrastive learning is impactful, as models like CLIP are now used widely, yet many undesirable behaviors such as gender biases still exist.

**Weaknesses:**

Significance: this paper relies on a preference dataset, which requires heavy annotations and the preference set will be very small compared to the training set of CLIP. Also, the preference would be very task-specific (e.g., typographic or gender), limiting the generalizability of the approach to new, unseen attacks or biases.

Quality: the inclusion of SVD makes it much slower to fine-tune on a larger scale. Also, the experiments focus on controlled, relatively smaller-scale datasets (the largest being ImageNet100), so the effectiveness of the approach is yet to be seen on diverse, complex large-scale datasets.

**Questions:**

Broadly speaking, for a general image-text task, e.g., VQA or retrieval, is there any guidance to design the preferences? The easiest way is following standard RLHF and curating a set with actual human preferences, but could the authors kindly suggest any other auxiliary information we can leverage?

---

> ### Author Response · Authors · 2024-11-24
> **Thanks for your careful review**
>
> Thank you for your great review and comments, they sparked some conversation in our team, and guided us to add some much needed clarifications to the paper.
>
> # About the weaknesses
>
> > This paper relies on a preference dataset, which requires heavy annotations and the preference set will be very small compared to the training set of CLIP. Also, the preference would be very task-specific (e.g., typographic or gender), limiting the generalizability of the approach to new, unseen attacks or biases.
>
> Thank you for raising this concern regarding the preference set. In Appendix I, we analyze the sample efficiency of our approach. Our study found that a relatively small dataset is sufficient to fine-tune the model to our preferences; for example, the dataset used for the gender reversal task consisted of approximately 2,000 sample images. This highlights the sample efficiency of our method, which does not require a large preference set.
>
> This optimization scheme applies to any task where there is a clear "preference". Simple examples include other targeted attacks, where an adversary aims to fool the model into outputting $y_l$ instead of $y_w$. Additionally, the method can address biases, such as harmful correlations amplified by the model, including racial bias, or more obscure cases, such as task bias, as discussed by [1].
>
> Regarding optimization, while our experiments focused on task-specific approaches, we believe that robustness against multiple types of attacks can be achieved by utilizing a more diverse preference set. This could improve generalization across various adversarial settings.
>
> As for multiple biases, the matrix $W$ will likely capture different biases through its singular vectors. However, the interpolation point for these biases may differ, in the gender bias setting, we found the interpolation point to be around $t=0.42$, reflecting how various biases are encoded across the singular value spectrum. One might try to use different matrices to disentangle this spectrum or find the most effective singular values regarding the specific bias they want to manipulate.
>
> However, we do acknowledge that untargetted attacks, and more generic, complex biases may be hard to address with our framework, since our framework works by comparing two different labels for each sample, this might be adressed by the KTO scheme since it does not require a preference dataset.
>
> > The inclusion of SVD makes it much slower to fine-tune on a larger scale. Also, the experiments focus on controlled, relatively smaller-scale datasets (the largest being ImageNet100), so the effectiveness of the approach is yet to be seen on diverse, complex large-scale datasets.
>
> Thank you for this comment. We have clarified this in newer version of the paper, the SVD is not included in the fine-tuning process. The fine-tuning procedure produces a learned matrix $W$, and the SVD is used as a post-processing technique for more fine-grained control over the learned features. It allows for adjusting and analyzing the learned matrix after fine-tuning rather than being part of the core optimization process.
>
> # On the questions
>
> > Broadly speaking, for a general image-text task, e.g., VQA or retrieval, is there any guidance to design the preferences? The easiest way is following standard RLHF and curating a set with actual human preferences, but could the authors kindly suggest any other auxiliary information we can leverage?
>
> While human-curated preference sets are ideal, our methodology can still be effectively applied by designing the pretraining task in a semi-supervised manner. In both of our experiments, we did not rely on publicly available preference sets or human-curated preferences. Instead, we designed them ourselves, tailored to the specific task we wanted to fine-tune on. For instance, in the first experiment, we generated synthetic typographic attacks and used the original and targeted labels as preference labels. For debiasing the model, we used a dataset of images showing individuals of each gender performing various tasks and fine-tuned the model on an auxiliary task of reversing the model’s gender understanding, again without curated preferences. This demonstrates that preference optimization can be effectively applied using task-specific, semi-supervised strategies.
>
> Since you have mentioned retrieval, in Figure 4 we showcase the results of our training on the retrieval task, where the model seems unbiased after the training process, even thought it was not exactly trained for retrieval.
>
> Previous work have used RLHF for many of the mentioned tasks, and exploring our methods capabilities (or even a preference optimization framework) in those tasks such as VQA could be a promising direction for future work.
>
> We have added this clarification to the paper.
>
> [1] Sachit Menon, Ishaan Preetam Chandratreya, and Carl Vondrick. Task Bias in Vision-
> Language Models, December 2022

---

> > ### Comment · Reviewer_PYYa · 2024-11-28
> > **Response**
> >
> > The reviewer thanks the authors for the response -- they addressed the concerns and the reviewer maintains the positive rating.

---

### Official Review · Reviewer_PfoD · 2024-11-04

**Soundness:** 3
**Presentation:** 4
**Contribution:** 2
**Rating:** 8
**Confidence:** 3

**Summary:**

This paper introduces an alignment method designed for contrastive models, such as CLIP, using aligned and unaligned image-text pairs. In this setup, each image input has a preferred (or aligned) response and an dispreferred (or unaligned) output. The model is trained to differentiate between these two responses using preference optimisation designed as a one-step Markov decision process. Therefore, they use a preference dataset that pairs images with aligned and unaligned responses, and a regularisation dataset containing clean examples to maintain the model's ability to generalise to other downstream tasks. Importantly, they don't fine-tune the full model but train a single linear projection layer on top of the frozen text and image encoders.

To further control the model's behaviour, the authors modify the singular values of the learned linear transformation. Specifically, they apply a singular value decomposition (SVD) to the weight matrix of this layer and scale all singular values using a scaling parameter $t$. This intervention technique builds on the intuition that the linear transformation transforms the original similarity function between image and text spaces.

They evaluate the effectiveness of their method in two settings. First, they evaluate its effect on typographic robustness by comparing it against baseline models (incl. standard CLIP, PAINT, Defense-Prefix) across nine datasets (incl. ImageNet, Flowers102, and EuroSAT). The preference dataset is created by adding misleading text to the original images of each dataset. They find that their method performs on par or better than prior methods, with a few exceptions. Despite improving the robustness, some performance gaps between the original and typographic dataset remain; for example, a gap of around 20 % on StanfordCars. Using the intervention technique leveraging the SVD of the linear projection layer, they show that they can modify the trade-off between OCR and object detection performance. In the second setting, they explore the possibility to disentangle gender representations. They train the linear transformation using a dataset of images depicting men and women during activities, and show that by scaling the singular values they can reverse gender-specific representations, including a specific scaling factor where the gender information is effectively neutralised, without significant degradation on the downstream task.

**Strengths:**

- This paper appears to be the first paper to apply preference optimisation to contrastive models, and presents an interesting use of SVD to control model behaviour.
- Optimising robustness and mitigating (gender) biases are of significant interest, especially in high-risk domains.
- The evaluation results suggest comparable and often better performance than alternative approaches in improving robustness while enabling a (to some degree) interpretable intervention technique.
- The paper is well written and easy-to-follow.

**Weaknesses:**

- Despite improving robustness over baseline methods in some datasets, none of the methods consistently outperforms other methods (see Table 1).
- The baseline methods, PAINT and Defense-prefix, and their differences to the proposed method are not explained in the paper.

Minor Comments:
- Line 23: Incomplete sentence „Our experiments We demonstrate“.
- Line 256: Comma instead of dot used.
- Line 258: Comma should be a dot, and dot should be a comma.
- Line 289: „this“ -> „This“
- The differences in Table 1 appear to computed inconsistently. While most of the time the differences are computed based on the best alternative method incl. the base model (e.g. OxfordPets), the difference for DTD O is computed with respect to PAINT, whereas CLIP seems to performs better. Overall, I think it would be easier to follow if all differences would be reported relative to the base CLIP model.

**Questions:**

- The role of the transformation scaling parameter $t$ in the results in Table 1 remains unclear to me. Is the parameter varied between all settings or kept constant?
- The scaling range from -2 to 1.2 in Figure 2 has likely been chosen because the performance improves up until that point. But what happens if you scale between e.g. -4 and 4? It would be interesting to see even if performance starts to deteriorate after some threshold.
- Have you tried training separate linear projection layers for the text and image decoders?

---

> ### Author Response · Authors · 2024-11-24
> **Thanks for your careful review**
>
> # 1. About the adressed weaknesses
> > Despite improving robustness over baseline methods in some datasets, none of the methods consistently outperforms other methods (see Table 1).
>
> Thank you for pointing this out, as it highlights an issue with our presentation of Table 1. The KTO variant of our model significantly outperforms all baselines on the typographic dataset and performs nearly identically to the state-of-the-art (SOTA) method on the original dataset. We have revised the table to make this distinction clearer.
>
> > The baseline methods, PAINT and Defense-prefix, and their differences to the proposed method are not explained in the paper.
>
> You are correct, this is due to our concerns about the page number limits, however, in the newer upload, we have added a short description of our baselines in the main text and a more detailed explanation in Appendix F
>
> # 2. About the minor comments
>
> Thank you for your thorough analysis. We have addressed these typos, and we agree that the last row is confusing. We have now modified it to be the difference between our best-performing model (KTO variant) and the best alternative method, excluding the original pre-trained CLIP, since it does not provide robustness. Let us know if you have any more suggestions.
>
> # 3. Answer to the questions
>
> > The role of the transformation scaling parameter $t$ in the results in Table 1 remains unclear to me. Is the parameter varied between all settings or kept constant?
>
> In Table 1, we did not vary the parameter, the table is the final result of training the model with each loss objective, without any tuning.
>
> > The scaling range from -2 to 1.2 in Figure 2 has likely been chosen because the performance improves up until that point. But what happens if you scale between e.g., -4 and 4? It would be interesting to see even if performance starts to deteriorate after some threshold.
>
> Thanks for the suggestion; We modified the plot and explained this behavior in the main text with extra analysis in Appendix E.
>
> This comment made us consider why we were not getting the best results beyond that point and led us to apply some design changes in section 3.4. To be exact, we just needed to normalize these embeddings when the transformation scaling is large. At small $t$, this normalization does not have much effect since $W$ is close to an orthogonal matrix; Therefore, it almost preserves length. After fixing this, the model has good behavior overall, with only some slight degradation at $t>3$, which is somewhat expected.
>
> > Have you tried training separate linear projection layers for the text and image decoders?
>
> That is a valid question, and we believe we should have explained our design choices. The choice to use a single projection layer was made for the following reasons.
>
> - **Interoperability**: While using separate projections, does work as a linear adaptation. Still, it does not provide the insights we had earlier, where our model would be amplifying or weakening certain aspects of the shared embedding space. It is still possible to make sense of what is happening; however, it is not as clear as in the case of a single projection. This would make interpolation harder, as the SVD does not hold the same intuition anymore. Using different matrices would treat the text and image embeddings differently.
>                 Currently, our method does the following:
>                 \begin{align*}
>                     \tilde{f}(y,x)&=\mathcal{I}(x)^\top V \Sigma^2 V^\top \mathcal{T}(y)=(\Sigma V^\top\mathcal{I}(x))^\top(\Sigma V^T\mathcal{T}(y))
>                 \end{align*}
>
>                 Therefore, at $t=1$, $\Sigma=I$ and our method becomes the identity, however, if we use different matrices, it would become the following:
>                 \begin{align*}
>                     \tilde{f}(y,x)&=(A\mathcal{I}(x))^\top(B\mathcal{T}(y))=\mathcal{I}(x)^\top A^\top B\mathcal{T}(y)=\mathcal{I}(x)^\top U \Sigma V^\top\mathcal{T}(y)
>                     =(\Sigma^{1/2}U^\top \mathcal{I}(x))^\top (\Sigma^{1/2}V^\top\mathcal{T}(y))\newline
>                     @ t=1&\Rightarrow \tilde{f}(y,x)=\mathcal{I}(x)^\top UV^\top\mathcal{T}(y)
>                 \end{align*}
>                 Therefore, even at $t=1$, we would have a different model and a slight rotational mismatch would be introduced.
>
>                 If we want the operator to become the Identity at $t=1$, we would need $U=V$, which, in effect, constrains us to the same search space as using a single projection layer.
> - **Parameter efficiency**: Using different matrices would effectively double the number of parameters.
> - **Consistency**: One of our previous baselines [1] also uses a single projection, by doing this, we are consistent with earlier work. They use implicit constraints on the orthogonality of $W$, where orthogonality is automatically preserved in our formulation.
>
> [1] Joanna Materzynska, Antonio Torralba, and David Bau. Disentangling visual and written concepts in clip, 2022

---

> ### Author Response · Authors · 2024-12-01
>
> Dear Reviewer PfoD,
>
> Thank you for your thoughtful review. We would greatly appreciate your feedback on our revised response. If you have any concerns, we would be happy to address them.

---

> > ### Comment · Reviewer_PfoD · 2024-12-02
> >
> > Thank you for your detailed response and specifically for the additional experiments provided in the global comment, in response to a question raised by reviewer y7hh; all of my concerns have been addressed. Overall, I appreciate the methodological contributions of the paper and believe that it could be a valuable contribution to ICLR. Therefore, I increase my score.

---

### Author Response · Authors · 2024-11-25
**Response to all reviewers**

We sincerely thank all reviewers for their careful reading and valuable comments. Here we address some common questions that are of interest to multiple reviewers.

The experimental results of our baseline methods were conducted by [1] on the ImageNetV2 validation subset, which was no longer publicly available at the time of our experiments. Instead, we evaluated our method on a different subset of the ImageNetV2 dataset, referred to as the "matched-frequency" subset. Table 4 provides a detailed comparison of our method with the pretrained CLIP model under the typographic attack on the ImageNet dataset.

Additionally, we have identified and addressed an issue in our presentation of Table 1. The KTO method significantly outperforms all baselines on the typographic dataset while performing nearly identically to the state-of-the-art (SOTA) method on the original dataset. We have revised Table 1 to make these distinctions clearer and provide a more accurate portrayal of the performance of our proposed methods across different datasets.

[1] Hiroki Azuma and Yusuke Matsui. Defense-prefix for preventing typographic attacks on clip. In Proceedings of the IEEE/CVF International Conference on Computer Vision

---

> ### Author Response · Authors · 2024-12-01
> **Analysis of effectiveness on other tasks**
>
> Dear all,
>
> Over the past week, we have conducted extensive experiments to evaluate our model's capabilities on diverse tasks. As a proof of concept, we tested our model for general adversarial robustness using PGD-based attacks on small test datasets.
>
> In this setup, we employed targeted adversarial attacks, where the adversary generates samples misclassified as $ y_l$ instead of $y_w$. The training process is online: adversarial samples are generated during training, with new samples crafted in each iteration to challenge the model. We leverage our DPO loss function's reference-free (ref-free) variant to enhance the model's robustness.  The regularization term becomes vital in this case, as the training samples are out-of-distribution, and typical adversarial training can lead to collapse. We utilized the reference-free variant of DPO to maximize performance since proximity on adversarial samples is not critical in this case.
>
> The final evaluation was performed using non-targeted PGD attacks with varying attack radii. Specifically, we utilized $\ell_\infty $-bounded attacks with $\epsilon = 2/255$ and $\epsilon = 4/255$. The results are summarized below:
>
> | Dataset   | Epsilon ($\epsilon$) | Train racc (%) | Train acc (%) | Eval racc (%) | Eval acc (%) | Pretrained acc(%) | Pretrained racc (%) |
> |-----------|-----------------------|----------------|---------------|---------------|--------------|--------------------|---------------------|
> | Eurosat   | 2/255                | 86.72          | 57.03         | 39.84         | 54.69        | 46.88             | 0                   |
> | CIFAR-10  | 2/255                | 87.11          | 98.04         | 62.89         | 98.43        | 91.79             | 0                   |
> | Eurosat   | 4/255                | 46.09          | 55.86         | 18.75         | 49.21        | 46.88             | 0                   |
> | CIFAR-10  | 4/255                | 62.50          | 96.87         | 21.09         | 96.48        | 91.79             | 0                   |
>
> Where acc and racc stand for accuracy on clean data and accuracy on adversarially perturbed data, respectively. The pretrained acc and  Pretrained racc columns signify the accuracy of the original model before any adversarial training.
>
> These results and further details of this experiment are included in Appendix K of the revised version of the paper. We plan to give more analysis in the final version. However, additional experiments will require more time due to the computationally intensive nature of adversarial robustness testing.

---

### Author Response · Authors · 2024-12-04
**Summary of the Rebuttal Period**

We sincerely thank all reviewers for their careful reading and valuable comments. During the rebuttal period, we addressed all typos, grammatical errors, and writing mistakes in the manuscript. We also carefully responded to reviewers' concerns and incorporated their feedback to improve the clarity and quality of the paper. Specifically, we revised several sections to resolve misunderstandings, ensuring the text is more accessible to a broader audience.

We improved the presentation of our main results in Table 1, highlighting the significant performance gains of our method compared to baseline approaches. The revised table now provides a more intuitive and clear report of our findings. Additionally, as suggested by reviewers, we expanded the explanations of baseline methods in Appendix F, ensuring readers have sufficient context for comparison.
Figure 2 was updated by increasing the scaling range for improved visualization. We also added a detailed explanation of this adjustment in the main text, complemented by further analysis in Appendix E.

We refined the definition of $\mathcal{L}_{\text{pref}}$ based on reviewer suggestions, ensuring it is precise and easy to understand. To strengthen the evaluation of our method, we included a detailed comparison of different variants of our proposed approach, which is now presented in Appendix J.

Additionally, we emphasized the comparative study design of our work, particularly highlighting the superior performance of Kahneman-Tversky Optimization (KTO). Updates were made to the abstract, contributions, and conclusion to better reflect this focus. To streamline the main text, some references were moved to the appendix as footnotes.

One reviewer (y7hh) raised an insightful question about the potential applicability of our method to tasks beyond enhancing robustness against typographic attacks and mitigating gender biases.

Inspired by this question, we conducted an additional experiment to test the versatility of our framework. Specifically, we evaluated our method's effectiveness in enhancing the robustness of the CLIP model against adversarial perturbations, such as PGD attacks. The results of this new experiment, along with implementation details, are provided in Appendix K, demonstrating the broader applicability of our framework. Unfortunately, we have not received further feedback from the reviewer regarding this addition.

Best, Authors.

---

### Meta-Review · Area_Chair_hCjD · 2024-12-21

**Metareview:**

The reviewers largely appreciated the paper's novel approach of applying Preference Optimization (PO) to enhance the CLIP contrastive learning model's robustness and reduce biases. Both Direct Preference Optimization (DPO) and Identity Preference Optimization (IPO) show promise in mitigating typographic attacks and bias. Evaluations on various datasets demonstrate comparable or superior results compared to existing methods. The method's clarity and novelty were widely praised, with some reviewers suggesting further elaboration on specific experiments and results. On the other hand, the manuscript would benefit from a clearer explanation of the trade-offs between methods like DPO, IPO, and KTO, as well as a more thorough discussion of their respective effectiveness in different settings. Furthermore, while the approach was seen as promising, several reviewers noted that the experiments primarily involved smaller datasets, which might not fully demonstrate the method's scalability to more complex, real-world tasks. Despite these critiques, the general consensus remained positive, with the paper seen as a valuable contribution to improving contrastive learning models. We ask the authors to incorporate the feedback and additional results from the discussion phase.

**Additional Comments On Reviewer Discussion:**

The work greatly benefited from the rebuttal phase.

---

### Decision · Program_Chairs · 2025-01-22

Accept (Poster)